

# Sources of Particulate Matter in the Athabasca Oil Sands Region: Investigation through a Comparison of Trace Element Measurement Methodologies

Catherine Phillips-Smith[1], Cheol-Heon Jeong[1], Robert M. Healy[2], Ewa Dabek-Zlotorzynska[3],
Valbona Celo[3], Jeffrey R. Brook[4], Greg Evans[1]

[1] Southern Ontario Centre for Atmospheric Aerosol Research, University of Toronto, Toronto, Ontario, Canada
[2] Analysis and Air Quality Section, Air Quality Research Division, Environment and Climate Change Canada, 335 River Road, Ontario, Canada
[3] Analysis and Air Quality Section, Air Quality Research Division, Environment and Climate Change Canada, 335 River Road, Ottawa, Ontario, Canada
[4] Air Quality Processes Research Section, Air Quality Research Division, Environment and Climate Change Canada, 4905 Dufferin Street, Toronto, Ontario, Canada

*Correspondence to:* Greg Evans (greg.evans@utoronto.ca)

**Abstract.** The province of Alberta, Canada is home to three oil sands regions which, combined, contain the third largest deposit of oil in the world. Of these, the Athabasca Oil Sands Region is the largest. As part of Environment and Climate Change Canada's program in support of the Joint Canada-Alberta Implementation Plan for Oil Sands Monitoring program, concentrations of trace metals in $PM_{2.5}$ were measured through two campaigns that involved different methodologies: a long-term filter campaign and a short term intensive campaign. In the long-term campaign, 24-hr filter samples were collected one-in-six days over a two-year period (Dec. 2010 - Nov. 2012) at three air monitoring stations in the Regional Municipality of Wood Buffalo. For the intensive campaign (Aug. 2013), hourly measurements were made with an on-line instrument at one air monitoring station; daily filter samples were also collected. The hourly and 24-h filter data were analysed individually using positive matrix factorization. Seven emission sources of $PM_{2.5}$ were thereby identified: two types of Upgrader Emissions, Soil, Haul Road Dust, Biomass Burning, and two sources of mixed origin. The Upgrader Emissions, Soil, and Haul Road Dust sources were identified through both the methodologies and both methodologies identified a mixed source, but these exhibited more differences than similarities. The second Upgrader Emissions and Biomass Burning sources were only resolved by the hourly and filter methodologies, respectively. The similarity of the receptor modeling results from the two methodologies provided reassurance as to the identity of the sources. Overall much of the $PM_{2.5}$ related metal was found to be anthropogenic, or at least to be aerosolized through anthropogenic activities. These emissions may in part explain



the previously reported higher levels of metals in snow, water, and biota samples collected near

the oil sands operations.





# 1    Introduction

The Athabasca Oil Sands Region, located in the north-east corner of the province, is the largest of the three oil sands deposits in Alberta, Canada (Bytnerowicz et al., 2010). This area contains an estimated 1.7 trillion barrels of oil, located tens of meters below the ground (Kean, 2009),

composed of a highly viscous mixture of high molecular weight hydrocarbons, bitumen, clay, sand, and water (Bytnerowicz et al., 2010). As of 2009, the oil was extracted at a rate of 0.825 million barrels per day (Moritis, 2010), predominantly through two methods: open pit mining and steam assisted gravity drainage (Canadian Association of Petroleum Producers, 2014). Combined, these methods have rendered 10% of the bitumen in the oil sands economically recoverable, making

Alberta home to the third largest known oil deposit in the world after Venezuela and Saudi Arabia (Xu and Bell, 2013).

The various processes involved in bitumen extraction are believed to have environmental impacts on the area's water (McMaster et al., 2006), soil (Whitfield et al., 2009), and ecology (Goff et al., 2013). While air quality studies are much more limited (Hodson, 2013; Bari and Kindzierski,

2015), gaseous emissions such as $SO_2$ and $NO_x$ are known pollutants associated with oil sands activities (Charpentier and Bergerson, 2009; Bytnerowicz et al., 2010; McLinden, 2012). These gases have been linked to several oil sands extraction processes such as mining, transportation, and upgrading (Howell et al., 2014). Of current interest are aerosol particles below 2.5 µm in diameter ($PM_{2.5}$), which affect the environment through transport of pollutants, visibility

reduction, and by directly or indirectly shifting the earth's radiation balance (Dusek et al., 2006; Posfai and Buseck, 2010; Jeong et al., 2013). Further, $PM_{2.5}$ has been linked to adverse health outcomes (Docker et al., 1993; Burnett et al., 1995; Schlesinger, 2007) due to its propensity to penetrate deep down to the alveolar region of the lungs (Borm and Kreyling, 2004; Alfoldy et al., 2009). $PM_{2.5}$ is produced both by natural and anthropogenic sources such as motor vehicles, wind-

blown dust, industrial processes, and biomass burning (Jeong et al., 2013). Past research on $PM_{2.5}$ within the Athabasca Region has included modelling (Cho et al., 2012), airborne studies (Howell et al., 2014), and comparisons of the regions $PM_{2.5}$ concentrations to other areas of Canada (Kindzierski and Bari, 2011; Kindzierski and Bari, 2012; Hsu and Clair, 2015).

No previous studies have examined short term variability in the metal composition of $PM_{2.5}$ in this

region. Compositional analysis of $PM_{2.5}$ can help elucidate sources and processes that contribute



to PM$_{2.5}$ mass concentrations. The metal species in PM$_{2.5}$ are of particular importance because they can be source-specific and are typically preserved in the aerosol phase during transport. For example, V and Ni are often indicative of oil combustion (Becagli et al., 2012), while Al, K, Mg, and Cr are indicative of road dust (Amato et al., 2014). This source specificity allows for the

identification of sources of PM$_{2.5}$ with great resolution (Moreno et al., 2009). Receptor models are often used to determine these sources in areas where the chemical composition of the various sources is unknown. One such receptor model is Positive Matrix Factorization (PMF), which uses a weighted multivariate statistical approach to identify pollution sources (called factors) by examining the correlations in the PM$_{2.5}$ metal speciation matrix over time (Paatero, 1996). In past

receptor modeling, open pit mining, upgrading, and fugitive dust have been identified as major emission factors in the oil sands region (Landis et al., 2012). However, these emission factors were identified based on metal species measured in lichen, which is not necessarily representative of PM$_{2.5}$.

Previous studies also provide indirect indications of higher levels of metals in this region. Through

dry and wet deposition, such as snowfall (Bari et al., 2014), it is possible for the metals contained in the particulate matter to reach the soil and surface waters in the area (Amodio et al., 2014). Metals, such as Cu, Zn, Ni, Cr, and Pb have been found to be higher in the Athabasca River, its tributaries, and snowpack near the oil sands developments than several hundred kilometers away (Kelly et al., 2010). Furthermore, epiphytic lichens have experienced increases in Ti, Al, Si, and

Ba (Landis et al., 2012). In summary, the available evidence suggests that metal contamination may already be occurring in this region and that some of this may be due to transport of metals present within PM$_{2.5}$.

Due to these gaps in knowledge, the purpose of this work was threefold: (1) To fill the knowledge gaps that exists about the sources of PM$_{2.5}$ metals in this region, (2) to assess the accuracy,

precision, and consistency of the Xact$^{TM}$ 625 instrument (Cooper Environmental Services, 2013) used for the intensive hourly measurements versus that of the more standard 24-hr filters, and (3) to determine what can be learned from receptor modelling using higher time-resolved vs. 24-hr filter data.

Since December 2010, under the Enhanced Deposition Component of the Joint Canada-Alberta

Implementation Plan for Oil Sands Monitoring (JOSM) Program, trace elements have been





measured by Environment and Climate Change Canada in PM$_{2.5}$ at three sites operated by the Wood Buffalo Environmental Association (WBEA), in close proximity to oil sands processing activities (Fig. 1). The 24-hr integrated filter samples are collected every 6 days (midnight-midnight) following the procedure used in the National Air Pollution Surveillance (NAPS) program. Furthermore, as part of the 2013 summer intensive field campaign, hourly measurements were made at one of the sites (Fort McKay South, AMS13) for one month (Aug. 10- Sept. 10) using a semi-continuous metal monitoring system; daily 23-hr filter measurements were also collected.

A comprehensive protocol was developed to analyze the data from the two methodologies individually with PMF, which made it possible to identify the sources of PM$_{2.5}$ affecting the measurement sites (Jeong et al., 2013; Sofowote et al., 2014; Jeong et al., 2016). The identity of these sources was supported by comparing the resolved source profiles with existing profiles for the postulated sources along with temporal patterns of measured gaseous species. Meteorological data (courtesy of the WBEA) was used to further improve interpretation and identify probable source locations. By comparing the PMF results of the 24-hr and hourly measurements, a deeper understanding of the long and short term temporal variability of the sources, and the applicability of the two measurement methodologies to receptor modelling was achieved.

## 2 Methods

### 2.1 Field Measurement Sites

Under the Air Component of JOSM, the Municipality of Wood Buffalo was selected for the monitoring of air pollutants associated with oil sands activities because it is home to both mining and in situ extraction operations. In the long-term filter study, metal concentrations were monitored around the Athabasca river valley at three WBEA air monitoring stations (AMS): AMS13- Fort McKay South (SYN); AMS5- Mannix (MAN); and AMS11- Lower Camp (LOW) (Fig. 1). The AMS13 site is located between three oil companies in the area, all of which perform extensive mining, upgrading, and in-situ processing (Fig. 1): Canadian Natural Resources Limited (CNRL) is to the north, Syncrude is to the south, and Suncor is to the southeast. All three companies extract bitumen through both open pit mining and in situ methods within the Athabasca Region. The other





two measurement sites are located farther south, directly between Syncrude and Suncor, with AMS11 to the north of AMS5.

Within the Municipality of Wood Buffalo, open pit mining is the predominant method of bitumen extraction. In open pit mining, large hydraulic shovels lift the bitumen-rich dirt into trucks for transport to a nearby wet crusher which reduces the size of the soil and adds water, allowing the soil-slurry to be piped to an upgrading facility (Syncrude Canada Ltd., 2015; Canadian Association of Petroleum Producers, 2014). Once at the upgrading facility the bitumen is separated from the slurry in large settling vessels, after which it is upgraded into different hydrocarbon streams using steam, vacuum distillation, fluid cokers, and hydrocrackers, these processes produce aerosol particles which are emitted to the air through the main upgrader stack (Landis et al., 2012). Other known sources of particles are: the large fleets of on and off-road vehicles, dust re-suspended by mining activities, evaporative emissions from tailings ponds, and dust re-suspended from open petroleum coke piles.

## 2.2    Instrumentation

### 2.2.1 Filter Monitoring Setup

$PM_{2.5}$ samples were collected at the three sites on 47 mm polytetrafluoroethylene (PTFE) membrane filters (Pall Corporation, New York) using Thermo Fisher Partisol 2000-FRM samplers at 16.7 L/min. The samplers were operated once every six days with a 24-hr sampling time (midnight-midnight) according to the NAPS protocol. All samples, including laboratory, travel, and field blanks, were subjected to gravimetric determination of PM mass and were subsequently analyzed for 22 elements using non-destructive x-ray fluorescence (ED-XRF). $PM_{2.5}$ samples were then analyzed for 37 trace elements including 14 lanthanoids by inductively-coupled plasma mass spectrometry (ICP-MS) combined with microwave-assisted acid digestion, which provides superior detectability for trace metal(oids) (Celo et al., 2011). The PMF analysis applied 2-yrs of filter data from Dec. 16, 2010 to Nov. 29, 2012 (Long-term Filter, Table 1).

### 2.2.2   Intensive Campaign Setup

During the intensive campaign in August, 2013, daily $PM_{2.5}$ samples were also collected at AMS13 site using a dichotomous sampler (Partisol 2000-D, Thermo Scientific, Waltham, MA) on 47 mm





PTFE filters (Pall Corporation, New York). The sampler was operated with a 23-hr sampling time (8:30 am – 7:30 am) so as to allow an hour for filter switching. In the dichotomous PM sampler, a virtual impactor splits the incoming $PM_{10}$ sample stream into fine ($PM_{2.5}$) and coarse ($PM_{10-2.5}$) fractions. Mass flow controllers maintained the flow rates of the fine and coarse particle streams

at 15 L/min and 1.7 L/min, respectively. Elemental composition of $PM_{2.5}$ was analyzed following the procedure described above. Due to the limited number of samples taken (sample number, n=29), these $PM_{2.5}$ data were combined with those of the long-term campaign for PMF analysis.

In addition to the filter measurements, an Xact 625 (Cooper Environmental Services, 2013) made hourly measurements of 23 metal species at AMS13 between August 10 and September 5, 2013

(n=489). This semi-continuous instrument was installed in a trailer and sampled air through a $PM_{10}$ head fitted with a $PM_{2.5}$ cyclone located 4.55 m above ground level. The Xact used a two-step "semi-continuous" process. In the first step, particles were pumped through a section of PTFE filter tape at a flow rate of 16.7 L/min, which was regulated through measurement of the inlet temperature and pressure. The section of filter tape was then analyzed in the second phase, which

employs the same measurement technique as ED-XRF. Both the sampling and the measurement phase occurred simultaneously, producing data for all 23 metals every hour.

## 2.3    Quality Assurance and Quality Control

The filter measurements were carried out in accordance with the standard operating protocols that

were in place and care was taken to ensure that quality assurance and control programs (ISO17025 accredited) were followed.

Quality assurance and quality control (QA/QC) for the Xact measurements was based on protocols implemented before, during, and after the intensive campaign. Prior to the intensive campaign, the Xact was calibrated using 12 high-concentration metal standards. Three metal standards: Cr, Pb,

and Cd, were selected to represent the three energy levels employed by the Xact. These metal standards were measured on site at the beginning of the intensive measurement campaign (Table S1). Throughout the campaign, the internal Pd, Cr, Pb, and Cd upscale values were recorded after the instrument's daily programmed test, and the $PM_{10}$ and $PM_{2.5}$ cyclones were cleaned weekly. A sample of filtered air was measured daily to determine both the detection limits (DL) and baseline



biases of the metals the instrument measured. Further QA/QC that occurred during the campaign can be found in the Supplementary. After the campaign, the performance of the Xact metals monitor was further evaluated through three methods: re-testing with the high-concentration standards, new medium-concentration standards, and a comparison of the Xact data to co-measured data from filter samples and other collocated high time resolution instruments (Supplementary S.1).

## 2.4     Data Analysis

### 2.4.1 Positive Matrix Factorization

The metal speciation data of the two measurement methods were analyzed using Positive Matrix Factorization (PMF). Developed by Paatero and Tapper, PMF is a least squares regression model that inputs the data (X) and uncertainty ($\sigma$) matrices of the receptor site, resolving them into factor profiles (F), factor contributions (G), and residuals (E) (Paatero and Tapper, 1993; 1994) (Equation 1). Each factor corresponds to pollution sources or processes that may co-occur, contributing to particles at receptor site; the profile displays the concentration of metal species within each factor, and the time series displays the normalized contribution of each factor to the total metal concentration over time (Norris and Duvall, 2014).

$$X = GF + E \qquad (1)$$

Marker elements within the factor profiles were identified based on their high concentrations and/or percent segregations. These marker elements enabled the initial attribution of these PMF factors to probable sources. The identity of these sources was then supported by comparing, where possible, the resolved PMF profiles with source profiles for the suspected sources along with temporal patterns of measured gaseous species.

Prior to running the PMF algorithm, the data were screened to exclude metals for which fewer than 10% of the measurements were above the detection limit (DL) (Table S2). The data for metals measured with both ED-XRF and ICP-MS were then compared so as to select the optimal set of measurements based on the percentage of data above the DL and the signal-to-noise ratio (S/N). For a full description of the PMF algorithm and pre-treatment, refer to Supplementary S.2. The diversity of the elements among the factors was examined using Shannon Entropy (Healy et al.,





2014) (Supplementary S.5). Elements with Shannon entropy above 3.5 were discounted as marker elements in the factor profiles due to their relatively equal segregation into the different factors.

The filter and Xact data were analyzed separately using PMF due to their different sampling intervals. Filter data from Dec. 2010-Nov. 2012 (Long-term Filter) and Aug. 2013 (Intensive

Filter) were combined (total n= 351) to produce a single data matrix as the Aug. 2013 data alone were insufficient to support a separate PMF analysis.  Measurements taken by ICP-MS were combined with those taken by ED-XRF in order to create a full metal profile. In instances where the ICP-MS and ED-XRF both measured the same species, the data measured by ICP-MS were selected as this resulted in the most above DL data. PMF solutions with four to six factors were

considered as candidates and five-factor solutions were selected for both the filters and the Xact data; these solutions had three common factors and two factors that differed (Figs. S5-S8). Furthermore, the five-factor solution for the combined filter data was similar to five-factor solutions produced when the filter data from each site was run independently (AMS5, AMS11, and AMS13) (Supplementary S.2).

**2.4.2   Supporting Analyses**

To further investigate the results, linear regression analyses were performed for each factor time series against NO, $NO_2$, $NO_x$, and $SO_2$ concentrations (obtained from the Wood Buffalo Environmental Association) in order to identify relationships. The time series of each factor resulting from both methodologies were run through a conditional probability function (CPF) to

determine the most likely direction of the relevant sources. As described in Equation 2, the CPF is the ratio between the number of times the mass contribution surpasses a certain threshold percentile (i.e. 75%) when the wind comes from a certain direction ($m_{\Delta\theta}$) and the number of times the wind came from that direction ($n_{\Delta\theta}$) (Kim and Hopke, 2004).

$$CPF = \frac{m_{\Delta\theta}}{n_{\Delta\theta}} \qquad (2)$$

In this study, the wind direction (obtained from the WBEA) was divided into 24 bins, each encompassing 15°, and time periods with wind speeds below 1 m/s were removed. Given the varied topography within the Athabasca River Basin, favouring transport of local emissions along the river valley, the CPF only gave approximate indications of the source directions. Finally, a back





trajectory model, Hybrid Single-Particle Lagrangian Integrated Trajectory (HYSPLIT) was run on potential non-local factors (Stein et al., 2015).

## 3 Results and Discussion

### 3.1 Elemental Species Overall Trends

Average metal concentrations from the filter data were compared to measurements taken by the NAPS program at seven different Canadian cities (Environment Canada, 2015) (Table S4). Prior to averaging, blank values were removed as described in Supplementary S.2, and Below Detection Limit (BDL) values were replaced by half the detection limit. Overall, the average concentrations of some of the metals measured through the long-term campaign were lower than those observed in the Canadian cities as shown in Table S4. This is not overly surprising as these cities are impacted by a range of anthropogenic activities such as heavy traffic and industrial factories. What was surprising was the number of metals measured in the largely unpopulated oil sands region, which exhibited similar or higher concentrations to those seen across the various cities. In particular, levels of Si, Ti, K, Fe, Ca, and Al appeared to be higher near the oil sands operations. However, these averages do not fully encapsulate the differences between the cities and the oil sands region. In the oil sands, large swaths of forest are broken up by the occasional mine or upgrader. When the wind comes from one of these directions, particularly the upgraders, there is a noticeable difference in the air quality. This is in contrast to a city where levels fluctuate less. To illustrate this large variability, the 90[th] percentile of the various metals was calculated and compared to the averages of the various cities. The results of this showed that at its highest peaks, the previously discussed elevated metals further surpass the average city values. Additionally, at the highest peaks the concentrations of S, Ba, Br, and Mn also surpass those typically seen in the cities.

### 3.2 PMF Results

Each technique identified five unique factors through PMF analysis. Comparison of these factors led to the identification of seven factors: two types of Upgrader Emissions, Soil, Haul Road Dust, Biomass Burning, and two factors of presumably mixed origins. Three of these factors were





identified by both methodologies, two by only one of the methodologies, and the two mixed sources showed more differences than similarities between the two methodologies. The results of the PMF factor profiles (F matrix) from the two methodologies can be seen in Fig. 2 (hourly data from the intensive campaign in Aug., 2013) and in Fig. 3 (24-hr filter data from the combined

long-term campaign, Dec. 2010 - Nov. 2012 and intensive filter campaign in Aug. 2013). Distinctive marker elements were evident in some profiles while other elements were surprisingly ubiquitous, appearing in most or all of the factors. Here the low diversity (high Shannon Entropy) of Ni and Se across the source profiles is interesting as it implies that these metal(oid)s are present in most of the sources, perhaps as a result of greater natural homogeneity in this region or

contamination of the region through anthropogenic activities.

### 3.2.1 Upgrader Emissions I

This factor was attributed to typical emissions from the upgrading processes based on the correlation (r=1.00, p<0.05 for the intensive campaign: r=1.00, p<0.05 for the long-term campaign) of its elemental profile with an average profile derived from samples of $PM_{2.5}$ taken

from main upgrader stacks in the area (Landis et al., 2012). Specifically, the elemental profile contained significant portions of the S, V, As, Br, and Pb (Fig. 3). The very high S contribution in its profiles distinguished it from the second upgrader related factor. The profile is suggestive of a mixed-combustion source as these characteristic elements have been previously seen in No.4 boiler fuels, a mix still used in many parts of Canada (Lee et al., 2000; Van et al., 2008). Further, the

high correlation of this factor with $SO_2$ (Table 2) was consistent with the increased $SO_2$ observed in fresh industrial plumes in the area (Hsu and Clair, 2015; Zhang et al., 2015). There were strong correlations in: i) the PMF factor profiles derived from the two methodologies and ii) the time series between the co-measured Xact and filter data of this factor (profile (r=1.00, p<0.05); time series (r=0.87, p<0.05)). These observations support the assertion that this factor resulted from

combustion of a range of fuels in support of upgrading processes.

### 3.2.2 Upgrader Emissions II

This factor was hypothesized to be a less common type of emissions originating from upgrading processes; it was only resolved through the hourly data of the intensive measurement campaign. More specifically, this factor was attributed to oil combustion because of the higher percentages

of V and Ni, (Fig. 2), which are typical of oil combustion (Huffman et al., 2000; Lee et al., 2000).



Other metals associated with oil combustion such as S, Ti, Zn, and Fe (Huffman et al., 2000; Lee et al., 2000) were also evident. The temporal correlation of this factor to $SO_2$ in the intensive measurement campaign suggested the combustion of high sulphur fuel (Table 2) (Van et al., 2008; Zhang et al., 2015). Correlation (r=0.90, p<0.05) between the chemical profile for this factor and the average metal profile seen in upgrader stacks around the region (Landis et al., 2012) was not as strong as for the Upgrader Emissions I factor. In addition, the time series of this factor exhibited short-term peaks that occurred during the intensive measurement campaign at different times than those of the first Upgrader Emissions factor. The lower mole ratio of particulate sulphur to $SO_2$ (Upgrader Emissions I: 0.44, Upgrader Emissions II: 0.22) evaluated during these short-term peaks suggests that while both sources were relatively local, Upgrader Emissions II may have been closer to the receptor site. This, in addition to the small contribution of this factor relative to that of the first Upgrader factor, indicated that this factor may have been due to occasional smaller plumes that occurred for too short a duration or too rarely to be differentiated through the long-term 24-hr filter data. In contrast, the Xact data's time resolution allowed isolation of this more-specific process/emission occurring as part of the upgrading processes. In this case the Upgrader Emissions factor for the long-term campaign likely included the Upgrader Emissions II factor. It is speculated that this factor may have arisen from short term changes in fuel, such as a switch to oil combustion for heat/energy or the burning of coke. A less likely, but more fortuitous, possibility is that the Upgrader Emissions II factor was due to a short-term change in upgrader fuel that occurred only during the intensive measurement campaign and not during the long-term campaign.

### 3.2.3 Soil

The Soil factor exhibited high concentrations of crustal elements such as Si, Ti, and K (Figs 2 and 3). Additionally, both the filter and the Xact's chemical profiles exhibited a high correlation with samples taken of the area's overburden dump (Intensive Measurement campaign (r=0.99, r<0.05); Filter (r=0.83, p<0.05)) (Landis et al., 2012) which supported identifying this factor as a soil factor. The overburden dump is comprised of the top soil of the area, a mixture of the soil and glacial till overlying the oil deposits (Landis et al., 2012). The Soil factor derived from the long-term campaign data exhibited high concentrations of additional crustal elements not measured by the Xact such as Al, and lanthanoids such as Pr, Sm, and Nd (Fig. 3). Interestingly, this factor also exhibited high concentrations of Fe (both campaigns) and S (intensive campaign). This may have been due to the presence of bitumen in the soil in the Athabasca region, or may indicate that this



natural crustal material was being aerosolized through anthropogenic means. Specifically, the soil may have been emitted through entrainment by off-road transportation, or crushing of bitumen-rich sand. This hypothesis was supported by the high correlation of this factor with $NO_2$ and $NO_x$ (Table 2), which are related to engine emissions (Almeida et al., 2014). In fact, it is probable that

this Soil factor represents a combination of emissions: (1) directly through entrainment by off-road vehicular traffic on-site and (2) indirectly through "track-out", where the dust temporarily sticks to vehicles travelling within the mining sites, only to be aerosolized on-road later after leaving the site. In summary, the Soil factor's chemical profile was consistent with natural soils but its rate of emission may have been enhanced by anthropogenic processes. This factor was identified through

both methodologies; the source profiles for the two campaigns and time series between the co-measured Xact and filter data were highly correlated (profile ($r=1.00$, $p<0.05$); time series ($r=0.87$, $p<0.05$)).

### 3.2.4 Haul Road Dust

Much like the Soil factor, this factor exhibited high concentrations of crustal elements such as Ca

(Fig. 2), and the PMF outputs derived from the intensive and long-term campaign data were highly correlated to each other (profile ($r=0.81$, $p<0.05$); time series ($r=0.79$, $p<0.05$)). What differentiated this factor from the Soil factor were the lower concentrations of Mn and Fe, as well as its lower correlations with both $NO_2$ and $NO_x$ (Fig. 2 and Table 2). Despite this, the strength of the correlations with $NO_2$ and $NO_x$ are indicative of vehicular traffic (Moreno et al., 2013; Almeida

et al., 2014). In fact, the source profiles for the Soil and Haul Road Dust were only weakly correlated ($r=0.15$, $p=0.629$). Furthermore, the metal profiles of this factor were similar to that of samples taken by WBEA of the Athabasca Region's haul road dust (intensive campaign ($r=0.93$, $r<0.05$); long-term campaign ($r=0.91$, $p<0.05$)). As haul roads are made of a mixture of overburden material combined with limestone and low-grade oil sand, it is reasonable that the Soil and Haul

Road Dust factors exhibit some similarities in chemical composition.

### 3.2.5 Biomass Burning

Only observed in the long-term campaign, this factor was characterized by its high S, K, Zn, Br, Cd, and Pb. All of these elements, to different degrees, have been associated with different types of biomass burning (Van et al., 2008; Vassura et al., 2014; Alves et al., 2011). Furthermore, this

combination of elements associated with different types of biomass combustion was consistent




with a forest fire (Landis et al., 2012) in which all types of plants are burned. Finally, this factor's profile displayed a high correlation (r=0.89, p<0.05) with the measured profile of an Alberta forest fire, and in fact experienced its highest peak during a period of intense forest fires in northern Alberta (USDA Forest Service, 2011). Combined, these similarities suggested that this factor

originated from biomass burning, with smaller possible contributions year-round from the scrap-brush burning for land clearing that is performed in the area.

### 3.2.6 Mixed Sources

The PMF results of each campaign yielded a factor that appeared to be a combination of anthropogenic and crustal sources. While the Mixed factors from the two campaigns were not

correlated to each other (profile (r=0.05, p>0.80); time series (r=0.04, p>0.80)), they both appeared to originate from multiple sources. The Mixed factor from the intensive measurement campaign was characterized by elements such as Cu, Zn, and Mn, which suggested the presence of mechanical abrasion (Zhang et al., 2011; Gietl et al., 2010; Bukowiecki et al., 2007), as well as K and Se, which suggested the inclusion of biomass or coal burning, perhaps the burning of scrap-

brush (Van et al., 2008; De Santiago et al., 2014). In contrast, the marker elements from the long-term campaign were Zn, Cu, as well as K, and Ca, suggesting as possible sources mechanical abrasion and activities aerosolizing crustal elements, respectively. The conflicting nature of these two different Mixed-source factors suggested the presence of further factors. This was further evidenced by the unstable temporal trends these factors exhibited and the relative stability of the

alternate 6-factor solutions (Supplementary S.3). For example, some of the characteristic elements identified in the intensive measurement campaign, such as Br and Se, proved to be less stable, and would separate from the intensive measurement campaign's Mixed factor when the number of factors was increased (Supplementary S.3). However, these 6-factor solutions were less stable than the 5-factor solutions, and did not yield additional, clearer factors. This suggested that these

Mixed factors were a result of insufficient data that prevented full separation into more distinct source profiles. The combination of anthropogenic and crustal metals present in these factors could have arisen from mining-related activities such as mechanical abrasion of excavated materials within crushers. Activities such as combustion powered mechanical abrasion could result in a full-range of PM, from ultrafine to coarse, part of which would be considered as $PM_{2.5}$ (Okuda et al.,

2007; Martins et al., 2015).





### 3.3    Spatial and Temporal Trends

After analysis with PMF, the resultant factor time series (G matrix) of the combined filter data was split into 3 distinct factor time series, one corresponding to each site (AMS5, AMS11, or AMS13), in order to examine both the temporal and spatial trends (Supplementary S.6). The time-series contributions were then averaged by to assess each site's seasonality (Fig. 4) from Dec. 2010 to Nov. 2012. Finally, the factor contributions derived from the Xact data were compared to the averaged contributions from the Aug. 2013 portion of the long-term campaign at AMS13 (Fig. 5).

As can be seen from Fig. 4, there were clear differences in mass contributions between the three sites. Of the three sites, AMS5 consistently exhibited the highest metal concentrations, followed by AMS11, then AMS13, which was largely due to the high amounts of Haul Road Dust measured at AMS5. In contrast, AMS11 had higher average contributions from the Upgrader Emissions and Soil than both AMS5 and AMS13. This can be explained by the prevalent winds at AMS11, which were often from the south-east, the direction of the upgrader processes (Fig. 1). Overall, these local differences between the sites may have been due to a combination of the proximity of the sites to the various emission sources, as well as the direction of the prevailing winds at each site (Supplementary S.12) In particular, the predominant winds at AMS13 came from the North and the Southeast, which point towards the Syncrude and the more distant CNRL mines. In contrast, at AMS5 the wind came from many directions in which there were roads and open mines.

Also Fig. 4 depicts the seasonal trends of the 5 factors identified in the long-term campaign. The total contribution was largest in the spring, then lower in the winter, then summer, then fall. A major part of this overall trend was due to the higher contribution from Biomass Burning, a natural source, in the spring. However, even without the inclusion of Biomass Burning, the combined mass loading of the 4 remaining factors followed the same overall seasonal trend. Of the 5 factors, only the Mixed factor was relatively consistent throughout the seasons (within 0.08 μg/m$^3$). Both the Soil and the Haul Road Dust factors were notably lower in the winter than in the other seasons, presumably due to freezing of the ground and snow cover in the winter. The concentrations were higher in spring than in fall. In the fall, the temperatures drop below 0°C quickly (Supplementary S.13) which could result in the lower Soil and Haul Road Dust concentrations. In contrast, temperatures are higher in the spring (Fig. S13) and there is a surplus of sand on paved roads from



ice treatment in the winter. Interestingly, the Upgrader Emissions factor's contribution to the total PM$_{2.5}$ mass was highest in the winter and spring, which may be due to the differences in mixing height and wind direction and speeds across the seasons, or it may reflect true changes in upgrader activity. Overall, the combined seasonal concentration of the factors follows the same trend as the strength and directionality of the prevailing winds, in which the most prevalent, fast winds are present in the spring, then winter, then summer, then fall (Fig. S12).

In addition to the wind speeds, mixing height plays a vital role in the impact of various factors throughout the seasons. In the colder months, inversions can occur, increasing concentrations and channeling emissions horizontally based on the local topography (Davison et al., 1981; Celo and Dabek-Zlotorzynska, 2010). This may result in higher contributions from emissions sources, such as Soil and Haul Road Dust, which occur close to the ground. In contrast, in the warmer months, more vertical mixing tends to occur, which may result in the observance of higher contributions from tall emission stacks. Thus, the seasonality apparent within the source contributions is presumably in part due to this seasonality in mixing. However, the low contributions of the Soil and Haul Road Dust factor in winter, when mixing is the lowest, is not likely due to low mixing, unless the mixing was so low that it prevents transport from the sources to the nearby measurement sites. This indicates it is more likely that emissions of Soil and Haul Road Dust are lower in winter.

A closer look at the contributions during the summer of 2013 is shown in Fig. 5. Good agreement of the Upgrader I and II Emissions can be seen between the PMF factor contributions to the total metals concentration calculated from the AMS13, Aug. 2013 filter and hourly data (53% and 54%, respectively). While there is disagreement between the two independent PMF runs in the magnitude of the contribution from Soil and Haul Road dust, the time series for the Soil and Haul road Dust factor from the filter measurements taken at AMS13 during Aug. 2013 vs. that from the Xact measurements did exhibit good temporal correlations (r>0.79). It is possible that the filter data only allowed limited separation of the Haul Road Dust factor from the Soil factor: the 24-hr time scale of the filter data would not enable distinguishing changes that occur on a smaller time scale, such as the changing mixing height and/or wind direction throughout the day. Alternatively, by combining the data from the three sites prior to running PMF, the factor contributions may have been biased towards sites more impacted by the crustal factors. Despite these differences in relative contributions, the factors identified by the two different methodologies were broadly similar.





Finally, as the filter data measured at AMS13 during Aug. 2013 was added to those measured between Dec. 2010 and Nov. 2012 prior to analyzing it with PMF, the very small contribution (~2%) of the Biomass Burning factor seen in this figure arose from the forest fire factor resolved through other times in the long-term campaign, and given its low magnitude it is uncertain if this

is an accurate indication of emissions from Biomass Burning during the intensive measurement campaign.

Whereas the long-term campaign was able to identify the overall seasonal trends of the factors, the lower time resolution made identification of some sources more difficult (Fig. S11). In contrast, the higher time resolution employed during the intensive measurement campaign revealed diurnal

patterns (Fig. 6 and Fig. S10). From Fig. S10, it is clear that both the Upgrader Emissions were observed episodically, which indicates that they likely came from specific point sources and could thus be measured only when the wind was favourable. Interestingly, both the Haul Road Dust and Soil exhibited similar, but slightly offset, diurnal trends (Fig. 6). This is in contrast to the daily trends of the mixing height in the area, which are highest during the day and lowest during the

night (Davies, 2012). This trend also disagrees with the average diurnal wind speeds that occur at AMS13, which experience their peak at 15:00, while both the Soil and the Haul Road Dust factors experienced their highest peaks at 11:00 (Fig. 6). Overall, this suggests that the daytime increases for both the Soil and the Haul Road Dust factors were not due to natural processes such as decreases in mixing height or increases in windblown dust. Further, the daytime increase pointed

to on-road vehicles (e.g. through track-out) rather than off-road mining activities as the more dominant source, as off-road operations at most mining sites occur around the clock.

In addition to diurnal patterns, the higher time resolution used by the Xact resulted in more detailed CPF plots (Fig. 7 and Fig. S14). Whereas the Xact data was able to point towards distinct emission sources with the CPF plots, the filter data time resolution was too low. Despite this, the multiple

sites used by the filter study allowed for approximate triangulation of the sources (Fig. S14). The results of the two CPF profiles indicate that both the Upgrader I and II Emissions came from the direction of known upgraders and that the Mixed factors came from the direction of known mines. The Haul Road Dust factor appeared to come from the direction of the three major roads closest to the receptor site(s). Further, the CPF profile of the Soil factor exhibited broad peaks which

surrounded known mines, perhaps as a result of vehicular traffic leading to and from the mines.



Finally, the Biomass Burning factor appeared to come into the valley from the north and south. Because of this, HYSPLIT analysis was run on the three highest-concentration days for both the Biomass Burning and the Soil factors in order to determine if they were local or regional in origin. The long-range winds of the Biomass Burning factor, which experienced its highest concentration

days during periods of known biomass burning just north of the measurement sites (USDA Forest Service, 2011), appeared to largely come from specific regions in northern Alberta that contained additional forest fires (Fig. S15). Combined, the local and regional forest fires greatly affected the air quality, as during this period (May-June, 2011), the average $PM_{2.5}$ concentration increased by 328%. Back trajectory analysis of the highest concentration soil days of each receptor site

suggested that it predominantly came from the direction of known mines (Fig. S16); these trajectories largely crossed uninhabited, forested, areas in northern Canada prior to reaching the open pit mines. Overall, these finding support the conclusions that the Upgrader Emissions came from the bitumen upgrading process, the Soil and the Haul Road Dust factors came from on road transportation coupled with "track out", and the Mixed factors likely originated in the open pit

mines.

### 3.4    Long-term *vs.* Intensive Campaign Methodologies

The methodologies employed in the long-term and the intensive campaigns each had their own strengths and weaknesses. The lower DL and higher number of metal species measured by the

filter samples resulted in more detailed factor profiles (Fig. 2 and Fig. 3). Further, the longer duration enabled identification of seasonal trends, while the use of multiple sampling sites improved geographic resolution of sources. However, the limited number of co-measured aerosol and gaseous species enabled fewer comparisons. Additionally, the longer 24-hr sampling time of the filter limited the separation of close proximity sources, such as Upgrader Emissions I and II.

In contrast, the higher time resolution employed by the Xact resulted in better defined temporal patterns which supported the separation of similar sources (Fig. 6 and Fig. S10), which in turn led to more precise CPF profiles and gaseous species comparisons (Fig. 7, Table 2). Furthermore, the higher time resolution measurements accomplished all this within a much shorter time frame. However, without the long-term filter sampling, it would have been unclear how representative

this intensive period was of the norm. Because of this, it is important to take more time-resolved



metals measurements in this area in order to see if there are further, unresolved sources. This is important as it can help to guide decisions made about regulation and control in the area.

In addition to their measurement capabilities, each campaign had its own requirements in terms of energy, difficulty in set up and measurement, and quality control. While the intensive measurements by the Xact had more energy and housing requirements on site, the filter analysis required much more follow-up laboratory analysis. Finally, despite the quality assurance and control measures employed by the Xact, comparison to co-measured species indicated a possible linear bias in certain metals measured. In contrast, the filter protocol employed was well established, and followed stringent quality assurance and quality control protocols. This led to a high level of confidence in these measurements.

### 3.5 Implications of the Source Identification for Element Species

Overall, the average concentration of some metal species measured in the Athabasca region was either equal to or higher than that measured in urban/industrial locations in Canada (Table S4). In particular, the concentrations of Si, Ti, K, Fe, Ca, and Al were, on average, higher than those measured in major Canadian cities. At their highest peaks the concentrations of S, Ba, Br, and Mn were also higher than the city averages. Of these metals, Ti, Fe, Cu, and Zn all showed periods of higher concentration during the intensive measurement campaign. Of these species, Al, Ca, Si, Mn, Ba, and Fe were predominantly observed in the Soil and Haul Road Dust factors, suggesting that they originated from vehicle-related emissions or associated anthropogenic dust production in the area. These species have been previously seen to be elevated in epiphytic lichens (Landis et al., 2012). The elements As, Pb, and Tl were associated with Upgrader Emissions, and Be and Ba with the Haul Road Dust; these metals and metalloids were also previously found to be higher in snow or biota near the oil sands operation (Landis et al., 2012).

There were some differences between the campaign data as to the dominant source(s) of some metals. For example, V was apportioned to Soil and Upgrader Emissions I in the long-term campaign while based on the intensive measurement campaign it was almost entirely apportioned to the Upgrader Emissions II factor. While this difference appeared to create some ambiguity it actually highlighted the enhanced separation of factors allowed by the higher time resolution Xact



data: V was associated with a type of anthropogenic emissions that the filter sampling had trouble identifying. Interestingly, Ni, Zn, Cr, Ag, and Cu were grouped into Mixed factors. As these factors represented a combination of multiple sources, the individual sources causing elevation of these metals is still not known; this limitation may help direct further studies including. More generally,

the metals used to create the factor profiles and thereby identify sources accounted for only a small portion of the total PM$_{2.5}$ mass. This limitation will be addressed in follow-up analysis combining the Xact data with other concurrent, time-resolved, measurements of non-refractory components. Combining these data will provide a more complete mass reconstruction so as to allow apportionment of PM$_{2.5}$ and further sources may be revealed by leveraging the perspective given

by the additional composition information.

## 4      Conclusions

In conjunction with JOSM, seven sources of PM$_{2.5}$ related metals were identified through two measurement technologies: two types of Upgrader Emissions, Soil, Haul Road Dust, Biomass

Burning, and two sources of mixed origin. Of the seven factors obtained by the PMF analysis, two were directly associated with oil sands upgrading, two with the transportation of the bitumen-rich oil, one with natural processes, and two with mixed anthropogenic-natural activities. Thus, much of the PM$_{2.5}$ related metals were found to originate from anthropogenic sources or activities. Interestingly, it was only through the time-resolved measurements taken by the Xact that some of

these anthropogenic activities became better defined and understood, which can help guide further studies. This work describes the influence of the development activities on PM in the part of the Athabasca Oil Sands Region near open pit mining and upgrading activities. Finally, determining the relative contributions of these sources to the different metals in PM$_{2.5}$, helped to better resolve their potential contributions to the higher concentrations of metals in snow, water, and biota that

have been previously reported for samples collected near the oil sands operations.



*Acknowledgements:* This study was undertaken with the financial and operational support of the Government of Canada through Environment and Climate Change Canada as part of the Joint Canada-Alberta Implementation Plan for Oil Sands Monitoring program. Infrastructure support was provided by the Canada Foundation for Innovation and the Ontario Research Fund (Project: 19606). The authors thank the Wood Buffalo Environmental Association (WBEA) for support in integrated air sampling collection in the Athabasca Oil Sands Region. We would like also to acknowledge the provincial, territorial and municipal governments as partners of the National Air Pollution Surveillance (NAPS) Program.





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





**Table 1. Summary of the measurement strategy used during the two campaigns.**

| Campaign | Sampling Interval | Monitoring Site | | |
| --- | --- | --- | --- | --- |
| | | AMS5 | AMS11 | AMS13 |
| Long-term Filter (Dec. 2010- Nov. 2012) | 24-hr Integrated Filters (one in six days) | ED-XRF /ICP-MS | ED-XRF /ICP-MS | ED-XRF /ICP-MS |
| Intensive Filter (Aug. 13- Sep. 10, 2013) | 23-hr Integrated Filters (daily) | *N/A* | *N/A* | ED-XRF /ICP-MS |
| Intensive (Aug. 10- Sep. 5. 2013) | 1-hr continuous | *N/A* | *N/A* | Xact metals monitor |

**Table 2. Pearson correlation (r) of gaseous pollutants with PMF-resolved factors.**

| Campaign | Factor | Correlated Gases (r>0.4, p<0.05) |
| --- | --- | --- |
| Long-term Filter | Upgrader Emissions I | $SO_2$ (r=0.52) |
| | Soil | none |
| | Haul Road Dust | none |
| | Mixed Sources | none |
| | Biomass Burning | none |
| Intensive | Upgrader Emissions I | $SO_2$ (r=0.84), $NO_2$ (r=0.44) |
| | Upgrader Emissions II | $SO_2$ (r=0.60), $NO_2$ (r=0.40) |
| | Soil | $NO_2$ (r=0. 63), $NO_x$ (r=0.54) |
| | Haul Road Dust | $NO_2$ (r= 0.61), $NO_x$ (r=0.49) |
| | Mixed Sources | none |





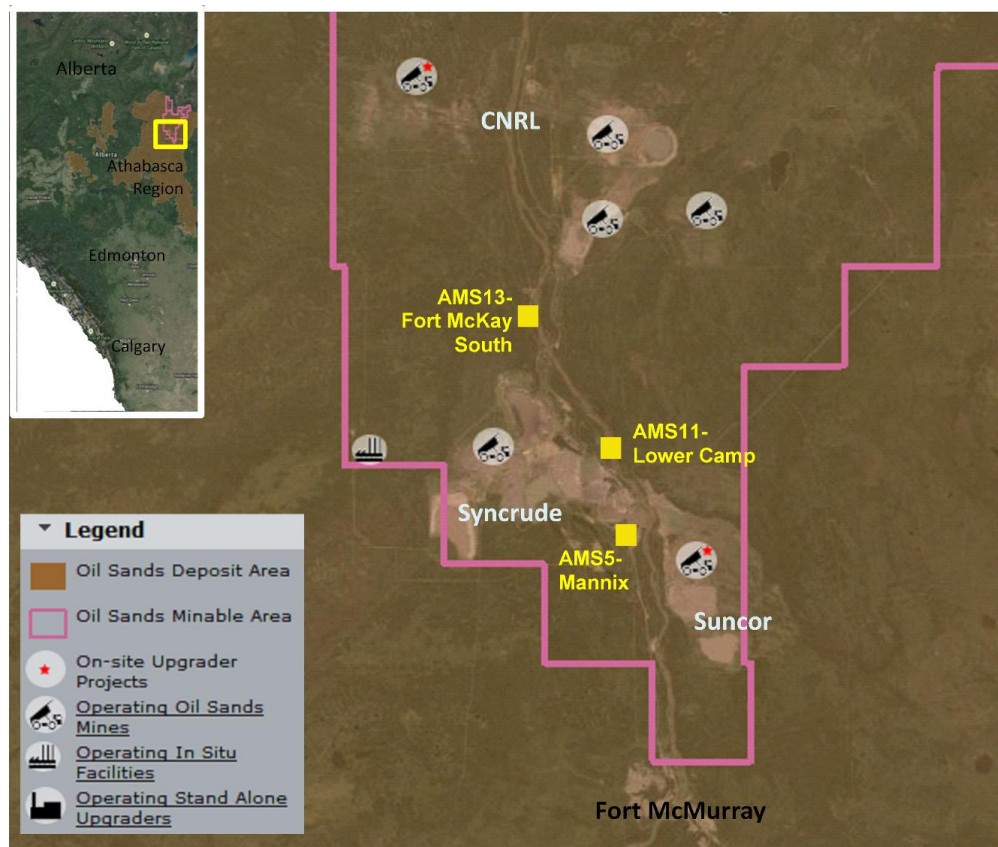

**Figure 1. Locations of the various extraction processes and three measurement sites (triangles) within the Municipality of Wood Buffalo in the Athabasca Region of Alberta, Canada. Map courtesy of Alberta: Environmental and Sustainable Resource Development. Available: http://osip.alberta.ca/map/**



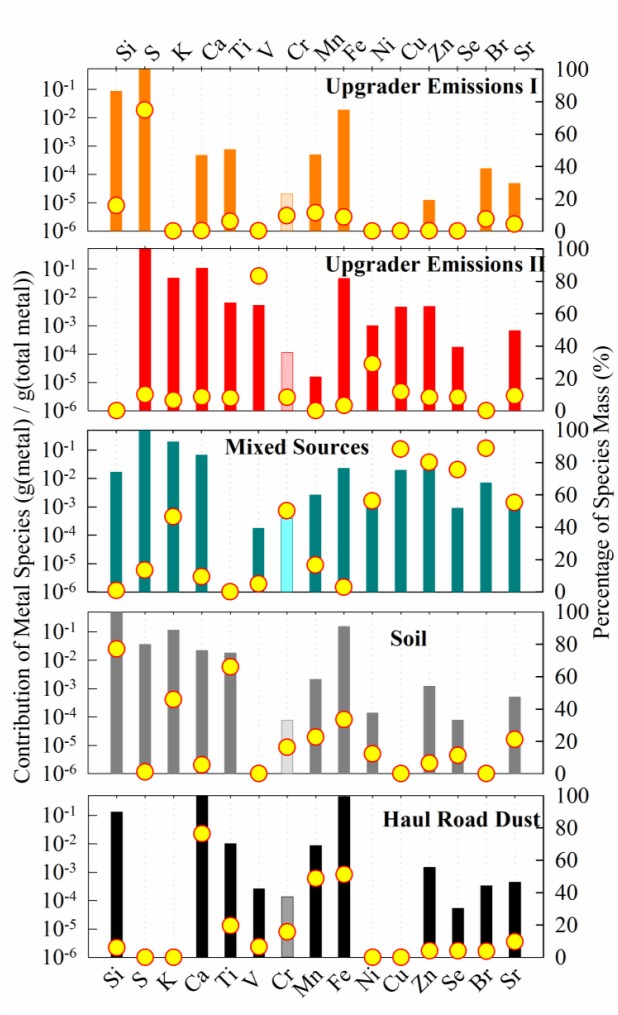

Figure 2. Factor profiles from the hourly measurements by the Xact, August 10- September 5, 2013. The percentage of species is defined as the percentage of mass of each metal apportioned to each factor. Elements with Shannon entropy below 3.5 have been given a higher transparency than the remaining elements.





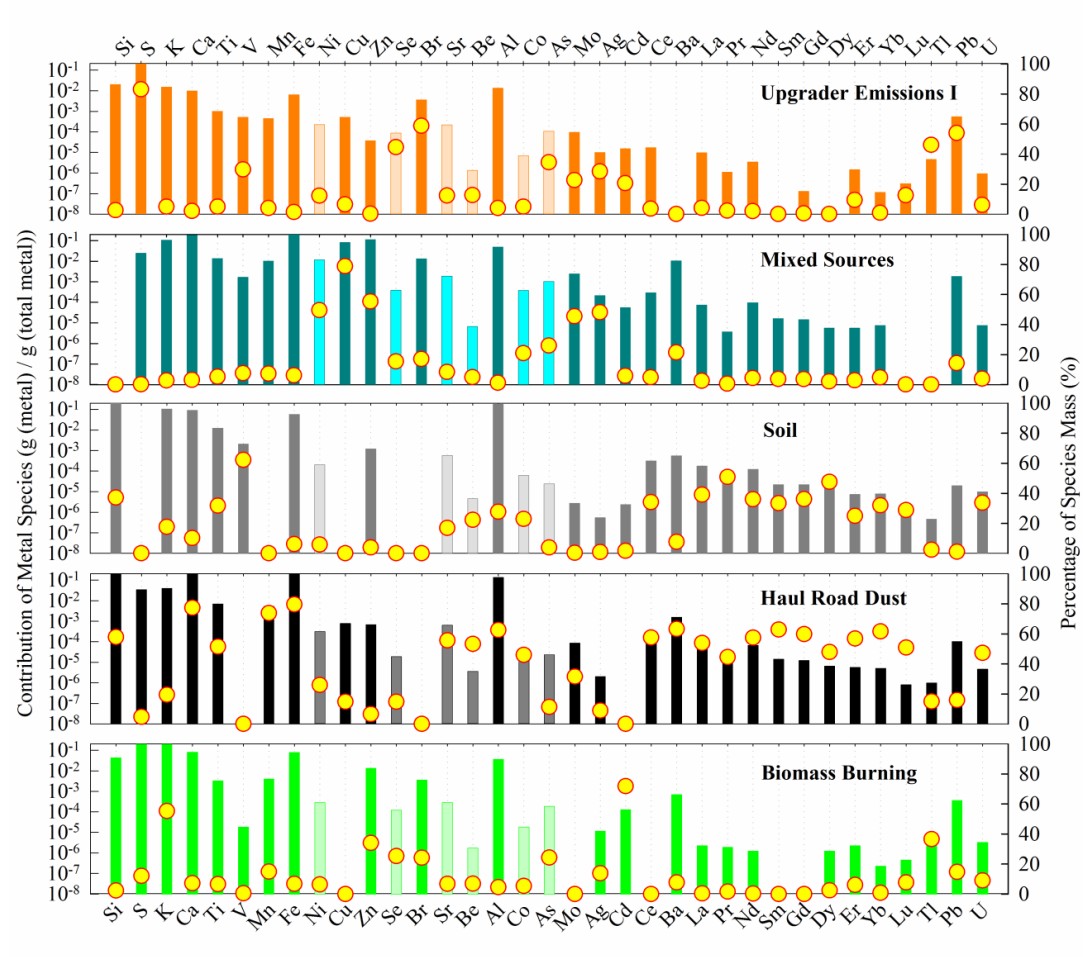

**Figure 3. Factor profiles of the combined filter data. Filters were collected one in six days at three sites for two years (Dec.2010-Nov. 2012) and daily at one site in Aug 2013. Elements with Shannon entropy below 3.5 have been given a higher transparency than the remaining elements.**





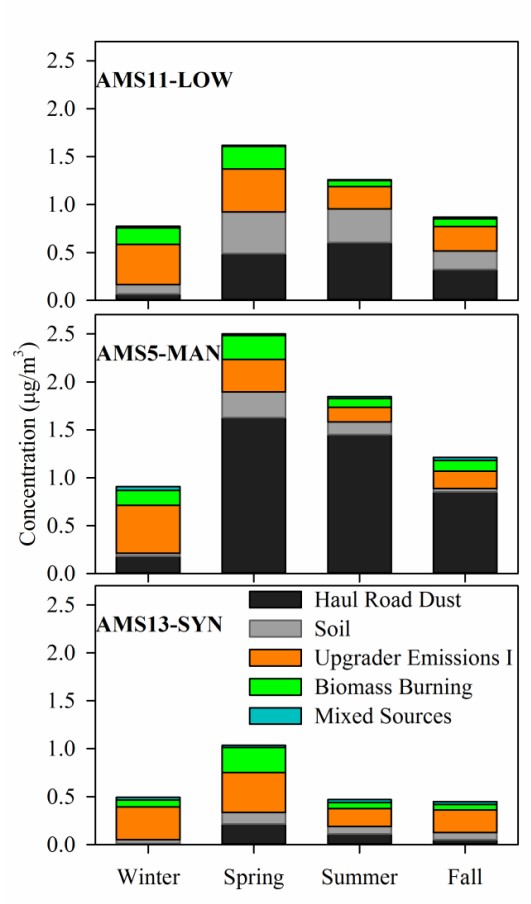

**Figure 4. Average seasonal contribution at the three sites from the combined long-term filter campaign data. The contributions are averaged by the season: winter (December 21- March 19), spring (March 20- June 20), summer (June 21-September 21), and fall (September 22-December 20)**
5    **from December 16, 2010 to November 29, 2012.**


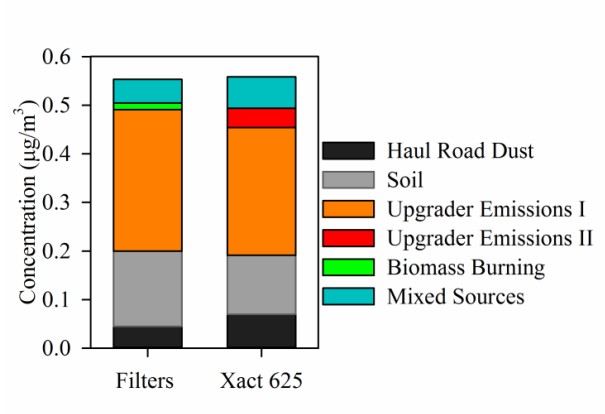

**Figure 5. Average mass contributions at AMS13 of each factor derived from the hourly Xact and 23-hr integrated filter data during the intensive campaign from August 13 to September 4, 2013.**

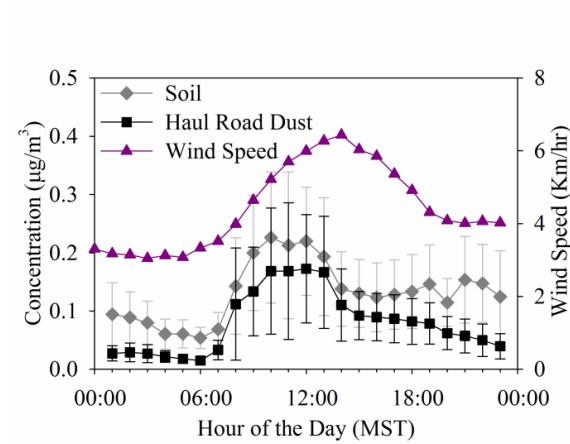

**Figure 6. Average concentrations displaying the diurnal trends the Soil factor (grey) and the Haul Road Dust factor (black) at AMS13 during the intensive campaign, and the average wind speeds between Dec., 2010 and Sept., 2012 (green). Error bars represent 95% confidence intervals.**

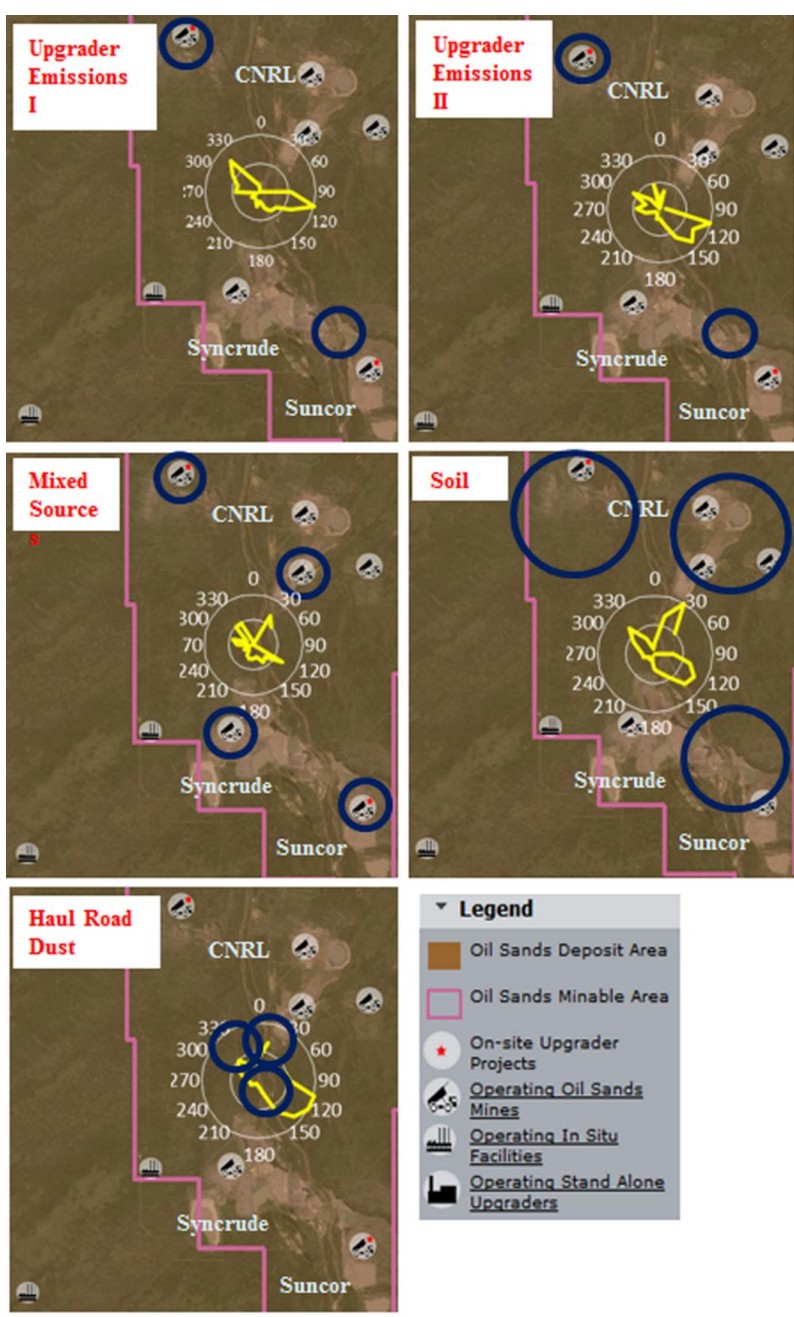

Figure 7. CPF profiles of the 5 factors identified by hourly Xact data during the intensive campaign
in August 2013. Blue circles indicate possible source locations. Map courtesy of Alberta:
Environmental and Sustainable Resource Development. Available: http://osip.alberta.ca/map/.