# Peer review of "Sources of Particulate Matter Components in the Athabasca Oil Sands Region: Investigation through a Comparison of Trace Element Measurement Methodologies"

_Atmospheric Chemistry and Physics, 2016_

## Referee Comment (RC1) · Anonymous Referee #1 · 16 Mar 2017

General Comments:

This manuscript describes an effort to apportion PM2.5 elements to different sources in the Athabasca Oil Sands Region. Two types of data were used: 24-hr filter samples from three near-mine monitoring stations over a two-year period from Dec. 2010 to Nov. 2012, and hourly elemental concentration from a portable elemental analyzer and 24-hr filters during an intensive study (Aug. 2013). Source apportionment was accomplished by Positive Matrix Factorization (PMF). Correlations with gases, time series, conditional probability function, and trajectory analyses were used to support

the rationality the PMF factors.

Elevated concentrations of elements have been observed in snow, river water, and lichen samples in the oil sands region (Bari et al., 2014; Graney et al., 2012; Huang et al., 2016; Kelly et al., 2010; Landis et al., 2012). Understanding the concentrations and sources of elements in PM2.5 in the region is important for assessing their environmental impacts and implementing control strategies. Therefore, the topic described in this manuscript is of great interest to the environmental science community, particularly to stakeholders in the oil sands region.

I have three major concerns on the data and results:

1. The source apportionment only used elemental composition. As can been seen from Figs. 2 and 3, many elements are present in multiple PMF factors, and most PMF factors are lacking specific elemental source markers. This collinearity make source identification and contribution apportionment less specific and more uncertain. The similarity in the temporal trend (e.g., Fig. S10) of different PMF factors is an indication of the difficulty in resolving these similar factors. The authors indicate that follow up analyses will include other chemical components (Page 20). I would prefer a source apportionment paper using all available chemical species (e.g., ion, carbon, isotopes, organic speciation, etc.) to create more specific and confident source apportionment results.

2. The Xact malfunctioned during the intensive study. While the authors tried their best to retain the Xact data as much as possible, the data quality is still in question for the following reasons:

a) There was a drop in the internal measurement values for Pd, Pb, Cr, and Cd between 8/25 and 9/2/2013. It was assumed that the regression Eqns in Fig. S1 were applicable to all metals within its energy level. However, Fig. S2 shows that such correction caused the slope of Xact sulfur vs. AMS sulfur to change from 2.75 to 3.57, a significant 30% difference. The authors speculate that "could have been due to an increase in sulfate size distribution" but did not provide such evidence. Did the dichotomous sampler coarse channel show higher sulfate after 8/25? b) Figs. S3d and S3e show that sulfur from filter and AIM-IC measurements were comparable (indicating non-sulfate sulfur may not be significant), but Xact was ∼40% higher. Such discrepancies also exist for several other high concentration elements (Fig. S3d), causing concerns about the Xact data quality. c) The Xact and filter data had various regression slopes according to concentration ranges (0.77 to 1.43; Figs. S3b-3d). From the assumption of using Fig. S1 to correct elements in the same energy level, I would expect the slopes to be similar for those within the same energy level, but not for those in the same concentration range. Both K and S belong to Energy Level 1 (Table S1), but the slopes are 0.77 and 1.43, respectively (Fig. S3c and 3d). This data is puzzling. Any explanations?

3. While trying to explain the results, there are quite a few hand waving speculations, which shows insufficient understanding of the potential pollution sources in the oil sands region. For example, the high sulfur in the stack flue gas was assumed to be caused by #4 boiler fuel instead of the sulfur origin from upgrading process. Several other speculations are commented later.

More specific and detailed comments are given below.

Specific Comments:

1. Title: "Source of Particulate Matter..." This is paper is about PM2.5 elements. Modify the title to be accurate.

2. The authors generally referred the elements as metal (e.g., Page 3 last paragraph), which is not accurate because there are non-metal elements (e.g., S and Si). I would suggest to call them element to be accurate.

3. There is a gap between the long-term filters and intensive study (Dec. 2012-July 2013). It would be logic to have the long-term sampling overlap the intensive study.

4. Page 3, 2nd paragraph. The review of past PM2.5 research in the oil sands is missing a large body of studies organized by the Wood Buffalo Environmental Association, such as air quality trend, emission source characterization, and elemental composition in lichen (Landis et al., 2012; Landis et al., 2017; Percy et al., 2012; Wang et al., 2012; 2015a; 2015b; 2016; Watson et al., 2012). These are very relevant to this manuscript.

5. Page 4 Line 3. "...V and Ni are often indicative of oil combustion..." While this statement might be true in general, it may not be accurate in the oil sands region because bitumen is enriched with V and Ni (Shotyk et al., 2016). Attributing V and Ni to oil combustion will not reflect the bitumen-rich environment of the oil sands region. Citation of marker species should consider the local sources and chemical nature in the study area. This also raises question about the reason for attributing the factor "Upgrader Emission to oil combustion due the higher percentage of V and Ni. This may not be due to oil combustion, but due to bitumen processing. Therefore, it is important to find out if oil is largely used in combustion in the mining/upgrading facilities.

6. Page 4 Line 3-4. "Al, K, Mg, and Cr are indicative of road dust..." This sentence was cited from a Mediterranean study, and the statement contradicts the results. For example, Al is abundant in almost all dust sources in the oil sands region (Wang et al., 2015a), K is a marker for biomass burning (Fig. 3) and Cr is not abundant in the haul road dust factor (Fig. 2).

7. In the quality assurance and quality control sections (maybe in corresponding supplemental materials), I suggest adding: a) A comparison of overlapping elements that were measured by both ICP-MS and XRF. b) Describe in more details about how PMF factors were optimized and their uncertainties were estimated (e.g., Landis et al., 2017; Reff et al., 2007). Each PFM factor should show uncertainty based on bootstrapping. The similarity of profiles may be examined with Chi-square in addition to correlation coefficients.

8. The calibration with metals standards showed good accuracy (Table S1) after the Xact returned to the laboratory after field campaign. Was there anything done to fix the

Pd signal drop during the field campaign? If so, what was cause of the problem and what was the fix? If not, did the instrument somehow fix by itself sometime between 8/25 and laboratory tests?

9. Page 11 Line 17-19. Please verify if No. 4 boiler fuels were used in upgrading facilities. From an earlier publication on oil sands stack emission (Wang et al., 2012), the boilers were fired with natural gas, process gas, and/or coke. Process gases (e.g., effluent from the sulfur recovery units) and coke, instead of No. 4 fuels, are likely the main sources of sulfur in that area. Careful survey and understanding of the industrial operations are required to reduce speculations.

10. Page 12. "It is speculated that this factor may have arisen from short term changes in fuel, such as a switch to oil combustion for heat/energy or the burning of coke..." This speculation of the source of "Upgrader Emission II" is not supported by hard evidence.

11. Sections 3.2.3 and 3.2.4. I am not convinced that the soil factor and haul road factor can be reliably separated based on elemental composition. The profiles Figs. 2-3 are very similar. The overburden itself is stable and is not a large source of fugitive dust (Wang et al., 2015b). Dust is emitted from dikes built by overburden when heavy haulers are travelling on them. Then the emissions are similar to the haul road. Emissions of haul road dust occur when heavy haulers are moving on the unpaved haul road and when wind speed is high. Since diesel-fueled heavy haulers are the largest emitters of NOx, I would expect the haul road dust to better correlate with NOx than the "Soil" factor. Large dust plumes are often observed from tailings beaches under high wind conditions. Also there are several unpaved roads around the sampling sites, which are large dust sources. As shown in Figure 6 and its related discussion, both Soil and Haul Road factors increased during the day with very similar temporal patterns and are not driven by wind speed. These indicate that these two factors are likely driven by road dust. A single "dust" factor instead two may be more appropriate.

12. Table 2. Have the correlations between PMF factors and CO been looked at?

CO is an indicator of biomass burning and/or vehicle emissions and may offer added evidence to the PMF factors.

13. Page 20 Line 16-17. It is not accurate to categorize the Soil and Haul Rod Dust factors as "two with the transportation of the bitumen-rich oil". This excludes on-road vehicle and other dust sources.

14. Figure 6 and S10. It might be useful to examine the diurnal variation each factor.

Technical Details

1. Figs 1, 7, S14 etc. All maps should have a scale bar to infer distances.

2. Figs. 2, 3, S5, and S6. Add a note to indicate if the bar and symbol refers to the left or right hand side of the Y-axis.

3. Page 6 Line 4. I suggest changing "dirt" to "oil sands".

4. Page 6 Lines 7-10. Each oil sands mining facility has multiple stacks that are connected to different upgrading processes. For example, Syncrude has a main stack, a FGD stack, and several smaller stacks. Furthermore, most stacks are equipped with pollution control devices. Some particles, e.g., ammonium sulfate, are formed in the FGD process designed to remove $SO_2$ (Wang et al., 2012). Rewrite this sentence to accurately reflect this information.

5. Page 6 Lines 10-13. Evaporative emissions from tailings ponds may be an importance source of VOCs and secondary particles, but may not be an important source for primary particles. Instead, windblown dust from tailings beaches and dikes is a significant particle source (Wang et al., 2015b).

6. Fig. S3b-S3d showed 9 metals. Where are other metals?

7. Write "Eqn 1" in supplemental materials in an equation form.

8. There are two Table S4

9. Page 10 Section 3.1 and Section 3.5. I don't think it is fair to compare the 90th percentile in the oil sands to the average city values. If such comparison should be done, the 90th percentile of city values should be used. Also the three monitoring stations are considered as near-source monitoring due to their close distance to mining facilities. It is not surprising that their elemental concentrations are higher than many cities, considering the abundance of dust in the region (Shotyk et al., 2016).

10. Page 13 Line 17. "lower concentrations of Mn and Fe" should be "higher…".

11. Page 20 Line 4. Incomplete sentence: including…

References:

Bari, M.A., Kindzierski, W.B., Cho, S. 2014. "A wintertime investigation of atmospheric deposition of metals and polycyclic aromatic hydrocarbons in the Athabasca Oil Sands Region, Canada." Sci. Total Environ. 485–486:180-192.

Graney, J.R., Landis, M.S., Krupa, S., Percy, K.E., (2012). Coupling lead isotopes and element concentrations in epiphytic lichens to track sources of air emissions in the Athabasca Oil Sands Region, Alberta Oil Sands: Energy, Industry, and the Environment. Elsevier, Amsterdam, The Netherlands, pp. 343-372.

Huang, R., McPhedran, K.N., Yang, L., Gamal El-Din, M. 2016. "Characterization and distribution of metal and nonmetal elements in the Alberta oil sands region of Canada." Chemosphere 147:218-229.

Kelly, E.N., Schindler, D.W., Hodson, P.V., Short, J.W., Radmanovich, R., Nielsen, C.C. 2010. "Oil sands development contributes elements toxic at low concentrations to the Athabasca River and its tributaries." Proceedings of the National Academy of Sciences 107 (37):16178-16183.

Landis, M.S., Pancras, J.P., Graney, J.R., Stevens, R.K., Percy, K.E., Krupa, S., (2012). Receptor modeling of epiphytic lichens to elucidate the sources and spatial distribution of inorganic air pollution in the Athabasca Oil Sands Region, Alberta Oil Sands: Energy,

[Figure]

Industry, and the Environment. Elsevier, Amsterdam, The Netherlands, pp. 427-467.

Landis, M.S., Patrick Pancras, J., Graney, J.R., White, E.M., Edgerton, E.S., Legge, A., Percy, K.E. 2017. "Source apportionment of ambient fine and coarse particulate matter at the Fort McKay community site, in the Athabasca Oil Sands Region, Alberta, Canada." Sci. Total Environ. 584–585:105-117.

Percy, K.E., Hansen, M.C., Dann, T., (2012). Air quality in the Athabasca Oil Sands Region 2011, Alberta Oil Sands: Energy, Industry, and the Environment. Elsevier, Amsterdam, The Netherlands, pp. 47-91.

Reff, A., Eberly, S.I., Bhave, P.V. 2007. "Receptor modeling of ambient particulate matter data using positive matrix factorization: Review of existing methods." J. Air Waste Manage. Assoc. 57 (2):146-154.

Shotyk, W., Bicalho, B., Cuss, C.W., Duke, M.J.M., Noernberg, T., Pelletier, R., Steinnes, E., Zaccone, C. 2016. "Dust is the dominant source of "heavy metals" to peat moss (Sphagnum fuscum) in the bogs of the Athabasca Bituminous Sands region of northern Alberta." Environ. Int. 92–93:494-506.

Wang, X., Watson, J.G., Chow, J.C., Kohl, S.D., Chen, L.-W.A., Sodeman, D.A., Legge, A.H., Percy, K.E., (2012). Measurement of real-world stack emissions with a dilution sampling system, Alberta Oil Sands: Energy, Industry, and the Environment. Elsevier Press, Amsterdam, The Netherlands, pp. 171-192.

Wang, X., Chow, J.C., Kohl, S.D., Percy, K.E., Legge, A.H., Watson, J.G. 2015a. "Characterization of PM2.5 and PM10 fugitive dust source profiles in the Athabasca Oil Sands Region." Journal of the Air & Waste Management Association 65 (12):1421-1433.

Wang, X., Chow, J.C., Kohl, S.D., Yatavelli, L.N.R., Percy, K.E., Legge, A.H., Watson, J.G. 2015b. "Wind erosion potential for fugitive dust sources in the Athabasca Oil Sands Region." Aeolian Res. 18:121-134.

Wang, X., Chow, J.C., Kohl, S.D., Percy, K.E., Legge, A.H., Watson, J.G. 2016. "Real-world emission factors for Caterpillar 797B heavy haulers during mining operations." Particuology 28:22-30.

Watson, J.G., Chow, J.C., Wang, X., Kohl, S.D., Chen, L.-W.A., Etyemezian, V., Percy, K.E., (2012). Overview of real-world emission characterization methods, Alberta Oil Sands: Energy, Industry, and the Environment. Elsevier Press, Amsterdam, The Netherlands, pp. 145-170.
* * *

---

## Short Comment (SC1) · 13 Apr 2017

In this manuscript the authors investigated sources of ambient concentrations of elements in fine particulate matter (PM2.5) at three industrial locations in the Athabasca Oil Sands Region (AOSR) using 24-h (Dec. 2010 – Nov. 2012) and 1-h (August 2013) data. The receptor model EPA PMF3.0 was applied and seven emission sources were identified. In general, the results appear to be impressive and interesting for the international scientific community. However, I would like to raise some points that would be needed to address to better understand and reveal the sources of PM2.5 in the AOSR.

[Figure]

Some assumptions and interpretations have been made particularly in the methodology and result sections, which make the findings more uncertain. I would therefore suggest that the authors should consider major revisions as outlined in the specific comments.

Specific comments: 1. The authors investigated sources of PM2.5 using trace element concentrations that accounted for only a small fraction of PM2.5 mass. The authors should consider using all available chemical components e.g., cations (K+, Ca2+), anions (SO42−, NO3−). It is reported that additional chemical composition data would be available in the follow-up analysis (page 20). To the best of my knowledge, carbonaceous aerosol (EC/OC) measurements were not performed at oil sands region. Therefore, checking PM2.5 mass closure is helpful to identify the proportion of unaccounted mass, which can be included as an input variable (missing mass) in the model as suggested by Larson et al (2006) and have been applied in several other studies (e.g., Wu et al., 2007; Bari and Kindzierski, 2017). This helps to better explain some source factors.

2. It is not clear how the authors come up with the 5-factor solution using EPA PMF3.0. The authors provided justification for choosing the optimum number of factors screening basic criteria e.g., Q-values, G-space plots, Fpeak values. However, they didn't report any error estimation techniques such as bootstrapping (BS) analysis. The authors should apply the current version of the PMF model (EPA PMF5.0) that allows to better assess the uncertainty of PMF solutions, using three error estimation methods i.e., BS, displacement (DISP) and bootstrapping with displacement (BS-DISP) analysis.

3. The selection of chemical elements for PMF analysis was based on the frequency of detection and species only >10% of the measurements above the detection limit were chosen. This may increase more uncertainty in PMF-resolved sources. It is suggested to exclude the elements with more than 70% of samples below the detection limit. The authors should include data quality information (including percentage of detection,

below detected and missing values) in the supplemental. It is also suggested to provide QA/QC for laboratory analysis.

4. The authors identified seven sources including two types of upgrader emissions, soil, haul road dust, biomass burning and two sources of mixed origin. It is suggested to improve the interpretation for describing some specific sources that are related to oil sands development. For example, adding secondary ions (SO42–, NO3–) in PMF analysis will help to better characterize the input of oil sands emissions.

5. Recent studies in the AOSR indicated fugitive dust (e.g., from oil sands tailings, unpaved roads and hauling fleet emissions) as a dominant source contributing $\sim$20%– 40% to PM2.5 (Landis et al., 2017, 2012; Bari and Kindzierski, 2017). The authors should check 'soil' and 'haul road dust' factors to better interpret the influence of fugitive dust.

6. The authors tried to compare the observed levels of PM2.5 elements in the industrial locations in the AOSR with other Canadian cities. Elemental levels at oil sands communities (e.g., Fort McKay and Fort McMurray) were not investigated here. Due to the unique nature of emission sources (not available in other Canadian region), the comparison analysis may not be informative and therefore suggested to exclude from the manuscript.

References: Larson, T.V., Covert, D.S., Kim, E., Elleman, R., Schreuder, A.B., Lumley, T., 2006. Combining size distribution and chemical species measurements into a multivariate receptor model of PM2.5. J. Geophys. Res. 111, D10S09, doi: 10.1029/2005JD006285.

Landis, M.S., Pancras, J.P., Graney, J.R., Stevens, R.K., Percy, K.E., Krupa, S., 2012. Receptor Modeling of Epiphytic Lichens to Elucidate the Sources and Spatial Distribution of Inorganic Air Pollution in the Athabasca Oil Sands Region. Alberta Oil Sands: Energy, Industry and the Environment, Elsevier, 427–468.

[Figure]

Landis, S.M., Pancras, J.P., Graney, J.R., White, E.M., Edgerton, E.S., Legge, A., Percy, K.E., 2017. Source apportionment of ambient fine and coarse particulate matter at the Fort McKay community site, in the Athabasca Oil Sands Region, Alberta, Canada. Sci. Total Environ. 584–585, 105–117.

Bari, M.A., Kindzierski, W.B, 2017. Ambient fine particulate matter (PM2.5) in Canadian oil sands communities: Levels, sources and potential human health risk. Sci. Total Environ. 595, 828–838.

Wu, C.-F., Larson, T.V., Wu, S.-Y., Williamson, J., Westberg, H.H., Liu, L.-J.S., 2007. Source apportionment of PM2.5 and selected hazardous air pollutants in Seattle. Sci. Total Environ. 386, 42–52.

---

## Referee Comment (RC2) · Anonymous Referee #3 · 16 May 2017

The manuscript by Phillips-Smith et al. investigated sources of PM2.5 in the Athabasca Oil Sands region based on PMF analyses of particulate metals. The metal species were derived from a long-term campaign in which filter-based measurements were conducted at three sites, and also from an intensive campaign in which semi-continuous measurements were performed at one of the long-term sites. Interestingly, PMF results were compared between these two campaigns. The topic of this manuscript is within the scope of the special issue "Atmospheric emissions from oil sands development and their transport, transformation and deposition". However, I cannot support its publication in the current form. As can be seen from my detailed comments given below, my
major concerns are about the PMF results.

1. Page 3, line 25-28. It is not appropriate to list "modelling", "airborne studies" and "comparison of PM2.5 concentrations" successively in one sentence.

2. Page 4, line 9-13. The authors implied that no source apportionment study had been performed for the oil sands region using metal species of PM2.5. However, it is unclear whether there are any previous source apportionment studies using other PM2.5 components (e.g., water-soluble ions, organic carbon, elemental carbon and etc.). Please clarify.

3. Page 4, line 24-26. No content in the results and discussion section corresponds to the second purpose presented here.

4. Line 29 in Page 4 to line 8 in page 5. This paragraph should be presented much more briefly, since all the descriptions involved here are repeated in the methods section.

5. Page 9, line 6-9. Please provide (representative) scatter plots comparing ICP-MS and ED-XRF measurement results on the same metals.

6. Page 11, line 12-15. It is unclear which PMF profile (i.e., Upgrader Emissions I shown in Figure 2 or 3) was used for the comparison to the profile reported by Landis et al. (2012). In addition, it is quite surprising that the regression analysis could show an r value of 1.00. Does this mean that the two profiles are exactly the same?

7. Page 11, line 29-30. V and Ni were used to indicate oil combustion. However, as shown in Figure 2, the majority of Ni was attributed to the Mixed Sources factor; on the other hand, negligible V was seen in the Mixed Sources factor. These results mean that the major sources of V and Ni are different. Consequently, I don't think it is reliable to attribute the Upgrader Emissions II factor to oil combustion, unless the authors could demonstrate that the V to Ni ratio calculated for this factor was comparable to that measured in source emissions from oil combustion.

8. Page 13, line 17. According to Figure 2 and 3, concentrations of Mn and Fe were

higher in the Haul Road Dust factor compared to the Soil factor.

9. Page 13, section 3.2.5. Figure 3 indicates that biomass burning was the major source of Cd. Moreover, the biomass burning contribution to Cd (∼80%) was more significant than its contribution to K (∼60%). However, previous source emission studies typically suggest that biomass burning is not a strong source of Cd (e.g., Schmidl et al., Atmos. Environ., 42, 126-141, 2008 and references therein). The authors are required to provide references to support their discussions here, i.e., biomass burning could be the major source of Cd.

10. Page 14, section 3.2.6. High abundance of Ni observed in the Mixed Sources factor should be discussed.

---

## Author Comment (AC1) · 27 Jun 2017

*General Comments: This manuscript describes an effort to apportion PM2.5 elements to different sources in the Athabasca Oil Sands Region. Two types of data were used: 24-hr filter samples from three near-mine monitoring stations over a two-year period from Dec. 2010 to Nov. 2012, and hourly elemental concentration from a portable elemental analyzer and 24-hr filters during an intensive study (Aug. 2013). Source apportionment was accomplished by Positive Matrix Factorization (PMF). Correlations with gases, time series, conditional probability function, and trajectory analyses were used to support the rationality the PMF factors. Elevated concentrations of elements have been observed in snow, river water, and lichen samples in the oil sands region (Bari et al., 2014; Graney et al., 2012; Huang et al., 2016; Kelly et al., 2010; Landis et al., 2012). Understanding the concentrations and sources of elements in PM2.5 in the region is important for assessing their environmental impacts and implementing control strategies. Therefore, the topic described in this manuscript is of great interest to the environmental science community, particularly to stakeholders in the oil sands region.*

*I have three major concerns on the data and results:*

1. *The source apportionment only used elemental composition. As can been seen from Figs. 2 and 3, many elements are present in multiple PMF factors, and most PMF factors are lacking specific elemental source markers. This collinearity make source identification and contribution apportionment less specific and more uncertain. The similarity in the temporal trend (e.g., Fig. S10) of different PMF factors is an indication of the difficulty in resolving these similar factors. The authors indicate that follow up analyses will include other chemical components (Page 20). I would prefer a source apportionment paper using all available chemical species (e.g., ion, carbon, isotopes, organic speciation, etc.) to create more specific and confident source apportionment results.*

We thank the reviewer for their helpful comments. The difficulty in adding other species lies in the variance of the sampling strategies and techniques between and within the two campaigns. In order to include as much data as possible, which increases the accuracy of the PMF analysis as it provides the program with more patterns to analyse, the filter data from all three sites and both campaign periods were included in the PMF run. However, only elemental composition data were available for these filters thus only trace element data was used for the high time-resolution study as well. As a main goal of the paper was to compare the source apportionment identified through a long-term, low time-resolution study and a short-term, high time-resolution study, the species analysed were kept as similar as possible. Follow-up research examining far more species would likely have to focus on the intensive period, when far more species were studied.

2. *The Xact malfunctioned during the intensive study. While the authors tried their best to retain the Xact data as much as possible, the data quality is still in question for the following reasons:*

a) *There was a drop in the internal measurement values for Pd, Pb, Cr, and Cd between 8/25 and 9/2/2013. It was assumed that the regression Eqns in Fig. S1 were applicable to all metals within its energy level. However, Fig. S2 shows that such correction caused the slope of Xact sulfur vs. AMS sulfur to change from 2.75 to 3.57, a significant 30% difference. The authors speculate that "could have been due to an increase in sulfate size distribution" but did not provide such evidence. Did the dichotomous sampler coarse channel show higher sulfate after 8/25?*

As stated in the Supplementary, estimated AMS PM1 sulfur and AIM-IC PM2.5 sulfur data were used to evaluate the instrument performance and the correction of Xact internal signals. We found more than 60% differences between AMS and AIM-IC sulfur data. Different cut-diameters and collection efficiencies of these two instruments could be a main reason for the difference. Thus we speculated that the change of the slope between AMS sulfur and Xact sulfur also might be related to a poor collection efficiency of the AMS as sulfate size distribution increases between near PM1 and PM2.5. The dichot sampler provides fine (PM2.5) and coarse mode (PM2.5-10) speciation data. However, the coarse mode sulfate data did not provide any stronger evidence of a shift in the sulfate size distribution between near PM1 and PM2.5.

Another possible explanation of the slope change before and after the Xact correction was identified and explored: the range of sulfur concentrations was quite different between the two periods. Before August 25, AMS sulfur increased to ~3500 ng/m$^3$, which was almost one order of magnitude higher than the sulfate range after August 25. In order to examine the effect of sulfur concentrations, AMS sulfur data before the 25$^{th}$ were regrouped into two ranges, < 300 ng/m$^3$ and > 300 ng/m$^3$. As shown in Figure R1a, the slope for the lower concentrations range was closer to the slope after Aug. 25 (3.26 vs. 3.57). At the high range the agreement between Xact and AMS sulfur was better (Figure R1b), which was close to the slope for the entire concentrations ranges (2.71 vs. 2.75). The reason of the concentration dependency is unclear, it could be due to the change of the Sulfate/Sulfur ratio varying by sources. However, the agreement of the Xact with the AMS is not the key point here as the agreement between these two instruments is still not well established. Rather, it is the similarity in the agreement of the Xact before and after the correction that is the point. Here the similar slope (~10% difference) for S before and after the Xact correction clearly supports the Xact correction is valid. Furthermore, Xact daily sulfur averages were compared to the filter XRF data before and after the correction. As shown in Figure R2, the corrected Xact S data lie in the range of the 95% prediction interval, indicating they are in the uncertainty levels of the analytical methods. In the revised supplement, Figure S2 was changed to display the slope at the lower range and we revised the statement on the comparison of the Xact data with SP-AMS.

"Overall, there was a large change in the slope of the line from 2.75 before August 25 to 3.57. However, the corrected slope was comparable to the slope in a similar concentration range (Figure S2b) before August 25 (3.26 vs. 3.57), suggesting that the Xact data correction was reasonable."

[Figure]

Figure R1. Comparison of SP-AMS sulphur equivalent to Xact S before Aug. 25, 2013 (a, b) and after Aug. 25, 2013 (c, d). All SP-AMS sulphate concentrations were divided by three to determine equivalent sulphur mass concentrations.

[Figure]

Figure R2. Comparison of daily integrated filter sulphur to Xact sulphur before and after August 25, 2013.

*b) Figs. S3d and S3e show that sulfur from filter and AIM-IC measurements were comparable (indicating non-sulfate sulfur may not be significant), but Xact was ~40% higher. Such discrepancies also exist for several other high concentration elements (Fig. S3d), causing concerns about the Xact data quality. c) The Xact and filter data had various regression slopes according to concentration ranges (0.77 to 1.43; Figs. S3b-3d). From the assumption of using Fig. S1 to correct elements in the same energy level, I would expect the slopes to be similar for those within the same energy level, but not for those in the same concentration range. Both K and S belong to Energy Level 1 (Table S1), but the slopes are 0.77 and 1.43, respectively (Fig. S3c and 3d). This data is puzzling. Any explanations?*

We agree that some hourly element concentrations measured by Xact were higher than the 23-hr integrated filter measurements. This difference is puzzling and thus we probed it in as many ways as we could and documented this in the supplementary. There could be many reasons for the discrepancy due to possible measurement biases of both methods (i.e., sampling time, inlet location, internal calibration factor). A further study for the comprehensive comparison of Xact and filter measurements is required to resolve confounding factors. In this study, the correlations between Xact and Filter for high concentration elements were very good ($r^2 > 0.85$), which was higher than the correlation between AIM-IC and Filter sulfur ($r^2 = 0.68$). We believe that this possible inaccuracy in the Xact data did not detract from the PMF analysis and the majority of elements included agreed well based on the filter comparison result.

The upscaling correction using Figure S1 equally affected all elements in the same energy level as the reviewer pointed out. However, each element in the Xact data has a specific internal calibration factor determined by standard element checks. Thus, the slope could be changed by elements and instruments.

3. *While trying to explain the results, there are quite a few hand waving speculations, which shows insufficient understanding of the potential pollution sources in the oil sands region. For example, the high sulfur in the stack flue gas was assumed to be caused by #4 boiler fuel instead of the sulfur origin from upgrading process. Several other speculations are commented later. More specific and detailed comments are given below.*

We have tightened up the language. In particular, we agree that Boiler Fuel #4 was added as an example of a potential source. This has been edited for clarity and accuracy as follows:

"The profile is suggestive of a mixed-combustion source (Lee et al., 2000; Van et al., 2008), such as coke, or the process gasses, which are comprised of effluent from the sulphur recovery units (Wang et al., 2012)."

*Specific Comments:*

1. *Title: "Source of Particulate Matter. . ." This is paper is about PM2.5 elements. Modify the title to be accurate.*

Since we didn't use major PM2.5 chemical speciation in PMF and multi linear regression of the source contributions against the total PM2.5 mass, the source apportionment and contributions are just related to the total metal mass, which are trace amounts within PM2.5. As such, the use of "Particulate Matter" encapsulates the purpose of the study for the title, as metals comprise such a small amount.

2. *The authors generally referred the elements as metal (e.g., Page 3 last paragraph), which is not accurate because there are non-metal elements (e.g., S and Si). I would suggest to call them element to be accurate.*

Agreed, and done

3. *There is a gap between the long-term filters and intensive study (Dec. 2012-July 2013). It would be logic to have the long-term sampling overlap the intensive study.*

It is agreed that it would be have been ideal to have a complete time series between the intensive and long-term campaign. However, at time of research and writing, these data were not available.

4. *Page 3, 2nd paragraph. The review of past PM2.5 research in the oil sands is missing a large body of studies organized by the Wood Buffalo Environmental Association, such as air quality trend, emission source characterization, and elemental composition in lichen (Landis et al., 2012; Landis et al., 2017; Percy et al., 2012; Wang et al., 2012; 2015a; 2015b; 2016; Watson et al., 2012). These are very relevant to this manuscript.*

It is agreed that the second paragraph of page three does not include a full account of all studies done in the area, as it is simply an introduction to some of the work done in the area, not a full synopsis. Some of these papers had been added to other parts of the paper, some have been added as per further comments. In order to provide a more full reference to the amount of research in the area, the section have been altered as follows:

"Past research on $PM_{2.5}$ within the Athabasca Region has included overall and comparative emission and air quality analyses (Kindzierski and Bari, 2011; Kindzierski and Bari, 2012; Wang et al., 2012; Percy et al., 2012; Howell et al., 2014; Wang et al., 2015; Landis et al., 2017). Further studies have developed into modelling the emission sources through both computer-based (Cho et al., 2012) and measurement-based methods (Landis et al., 2012)"

5. *Page 4 Line 3. ". . .V and Ni are often indicative of oil combustion. . ." While this statement might be true in general, it may not be accurate in the oil sands region because bitumen is enriched with V and Ni (Shotyk et al., 2016). Attributing V and Ni to oil combustion will not reflect the bitumen-rich environment of the oil sands region. Citation of marker species should consider the local sources and chemical nature in the study area. This also raises question about the reason for attributing the factor "Upgrader Emission to oil combustion due the higher percentage of V and Ni. This may not be due to oil combustion, but due to bitumen processing. Therefore, it is important to find out if oil is largely used in combustion in the mining/upgrading facilities.*

This paragraph is intended as an example of how metals can and have been grouped together for source apportionment in the past, not as evidence of results. As bitumen is an oil derivative, this similarity would not be unexpected. Paragraph reworded to elucidate:

"However, compositional analysis of $PM_{2.5}$ has helped elucidate sources and processes that contribute to $PM_{2.5}$ mass concentrations in other regions. The element species in $PM_{2.5}$ are of particular importance because they can be source-specific and are typically preserved in the aerosol phase during transport. For

example, V and Ni are often indicative of oil combustion (Becagli et al., 2012), as well as oil derivatives (Shotyk et al., 2016),"

6. *Page 4 Line 3-4. "Al, K, Mg, and Cr are indicative of road dust. . ." This sentence was cited from a Mediterranean study, and the statement contradicts the results. For example, Al is abundant in almost all dust sources in the oil sands region (Wang et al., 2015a), K is a marker for biomass burning (Fig. 3) and Cr is not abundant in the haul road dust factor (Fig. 2).*

It is agreed that this is a source from a Mediterranean study. The purpose of this paragraph is to introduce how metals have been grouped together in the past for source apportionment, this has been edited for clarity as follows:

"while Al, Mg, and Cr, when grouped together have been indicative of dust, in the past specifically that associated with transportation (Amato et al., 2014)"

7. *In the quality assurance and quality control sections (maybe in corresponding supplemental materials), I suggest adding: a) A comparison of overlapping elements that were measured by both ICP-MS and XRF. b) Describe in more details about how PMF factors were optimized and their uncertainties were estimated (e.g., Landis et al., 2017; Reff et al., 2007). Each PFM factor should show uncertainty based on bootstrapping. The similarity of profiles may be examined with Chi-square in addition to correlation coefficients.*

Scatter plots for Al, Ti, V, Mn, Fe, and Zn measured by ICP-MS and ED-XRF have been added in the supplementary. The correlation coefficients ($r^2$) ranged from 0.81 to 0.96 with good agreements.

[Figure]

Figure R3. Comparison of ED-XRF and ICP-MS measurements at AMS 5, 11, and 13 from Dec. 16, 2010 to Nov. 29, 2012. Nine elements with more than 80% of data below the minimum detection limit were excluded from the comparison.

In the revised manuscript, we re-examined the PMF solution using EPAPMF 5 and calculated comprehensive error estimates of the PMF solution. Possible solutions were compared to determine the best solution based on their stabilities and uncertainties. Detailed information about the solution evaluation has been added in supplementary as follows:

"In order to estimate uncertainties and evaluate the robustness and rotational ambiguity of PMF modeling results, the solutions were evaluated using the error estimation methods of EPA PMF 5; bootstrap analysis (BS), displacement analysis (DISP), and bootstrap enhanced by displacement (BS-DISP). Bootstrap analysis (BS) was performed to quantify the uncertainty of a PMF-resolved solution. In addition, 100 bootstrap iterations were conducted to obtain the percentage of factors assigned to each base case factor (i.e. bootstrap mapping) and determine unstable factors in the PMF solutions. With the displacement analysis (DISP), each element in source profile is displaced from it fitted value in a PMF solution to estimate the uncertainties for each element in each factor profile. Based on the result of the displacement analysis of a PMF solution, the rotational ambiguity of PMF solutions was assessed (i.e. number of swaps at the lowest predetermined Q levels). BS-DISP, a combination of BS and DISP, estimates the error associated with both random and rotational ambiguity (Paatero et al., 2014; Brown et

al., 2015). A discussion of diagnostic results of the error estimation methods for possible PMF solutions is provided in Supplement S.3."

"The bootstrap (BS) analysis was conducted to evaluate the uncertainties (i.e. random error in data values) of the source profiles and the reproducibility of factors in every bootstrap (Paatero et al., 2014; Brown et al., 2015). In the BS analysis, the BS factors are compared with the base run factors and then mapped to the base factor if the correlation is higher than a threshold ($r^2$=0.8 in this study). Tables S4 and S5 summarize the diagnostics of the error estimation for three PMF solutions (i.e. 4-, 5-, 6-factor solution) for the intensive Xact data and the long-term filter data, respectively.  In the 5-factor solution of both Xact and filter data, we found most bootstrap factors were well assigned to base factors in >96% of every bootstrap. Overall reproducibility (i.e. average BS mapping percentages) for each factor in the 5-factor solution was higher than other solutions, suggesting the 5-factor solution was very reproducible and the optimal solution. The displacement (DISP) analysis was conducted to evaluate rotational ambiguity in the PMF solution as well. Multiple solutions may be generated with the same value of the object function Q due to rotational ambiguity. In DISP, each fitted element (only good species) in a source profile is displaced in turn from its fitted value until Q increases by a predetermined maximum change in Q. An uncertainty estimate for each element in each factor profile is thereby yielded and factor swaps may occur if factors change too much. A comprehensive error estimate method, bootstrap enhanced by displacement (BS-DISP) combine the strengths of BS and DISP, which evaluate both the robustness to data errors and rotational uncertainty. Overall, no change in DISP Q (%dQ) was found for the 5-factor solutions. Furthermore, no swapped factor was found in DISP BS-DISP runs, indicating the 5-factor solution was a global minimum and well-defined PMF solution."

"The source profiles of the 4- and 6-factor solution for the intensive Xact PMF analysis are shown in Figures S5 and S6. In the 4-factor solution, the Soil and Haul Road Dust factors can be combined, but the reproducibility of the solution was poor and there were factor swaps in the BS-DISP runs. In the 6-factor solution, the Soil factor from the 5-factor solution split into two similar soil factors which have poor BS mapping reproducibility and very high factor swaps in the BS-DISP analysis. Another solution could be possible in the 6-factor solution which was characterized by additional resolution of the Mixed sources in the 5-factor solution. As shown in Figure S6b, two Mixed sources were characterized by high loadings of Cu (Mixed Sources I) and Br and Se (Mixed Sources II). Thus, due to the robustness of the solution and the physically meaningfulness of the factor profiles, the 5-factor solution was clearly acceptable for the intensive campaign."

"In the long-term campaign the Haul Road Dust and Soil factors in the 5-factor solution can be combined into one factor in the 4-factor solution (Figure S7a). However, there was an alternative solution including a combined factor of Mixed and Upgrader Emissions (Figure S7b). Due to the instability, there were factor swaps in BS-DISP and the reproducibility of the 4-factor solution was poor. With 6 factors, an additional Soil factor characterized by high loadings of rare earth elements and vanadium was found. However, this second Soil factor is only found 56% of the BS resamples and 85% of the BS-DISP runs were accepted with high factor swaps. In the 7-factor solution, additional factor containing high Pb and Br can be isolated from the Upgrader Emissions factor, but it's stability was very poor and there was no reasonable source for Pb and Br only. The BS resamples and BS-DISP runs of the 5-factor solution was better than the 4- and 6-factor solutions for the long-term filter data (Table S5). These results indicate that the 4- and 6-factor solution are much less certain than the 5-factor solution. As a result, the 5-facror solution was chosen as the most reasonable and stable solution for the filter data."

"In order to determine the relative weights of the different factors, a multiple linear regression of the time series of each factor for both campaigns was run against both the summed metal concentrations as well as

the PM2.5 concentrations (obtained from WBEA). As trace metals only account for a small percentage of the overall PM2.5 mass, the results of the PM2.5 regression proved to be a poor fit both statistically ($r^2$<0.8) and physically, as it resulted in negative relative weights, to the metal speciation factor solutions. Because of this, the total metals concentration was used to determine the relative weights of the different factors, which resulted in a much better fit ($r^2$>0.99)."

The chi-square value is the weighted sum of squares of the differences between the measured (i.e., Landis et al. 2012) and PMF-modelled element concentrations. However, no Chi-square values are available in this study since the Landis's source profiles were reported as relative contributions (i.e., µg/g PM). In the revised manuscript, uncentered correlation coefficient has been used as an unbiased metric to evaluate the level of similarity between the profiles of sources. This metric is particularly useful as it takes into account the similarity of minor peaks.

**Table S4. Summary of error estimation diagnostics for intensive Xact data.**

| | 4-Factor Solution | 5-Factor Solution | 6-Factor Solution |
|---|---|---|---|
| Robust Mode | Yes | | |
| Seed Value | Random | | |
| # of Bootstraps in BS | 100 | | |
| $R^2$ in BS | 0.8 | | |
| DISP active species | Si, S, K, Ca, Ti, V, Mn, Fe | | |
| BS-DISP active species | S, K, Ti, V, Fe | | |
| Factors with BS mapping < 100% | Upgrader Emissions II(51%), Mixed (99%) | Upgrader Emissions II (98%) | Soil (53%), Soil II (93%) |
| DISP %dQ | 0 | 0 | 0 |
| DISP # of swaps | 0 | 0 | 0 |
| BS-DISP % of Cases Accepted | 95 | 97 | 27 |
| BS-DISP # of swaps | 4 | 0 | 198 |

**Table S5. Summary of error estimation diagnostics for long-term combined filter data.**

| | 4-Factor Solution | 5-Factor Solution | 6-Factor Solution |
|---|---|---|---|
| Robust Mode | Yes | | |
| Seed Value | Random | | |
| # of Bootstraps in BS | 100 | | |
| $R^2$ in BS | 0.8 | | |
| DISP active species | Si, S, K, Ca, Ti, Fe, Cu, Sr, Al, Cd, Ce, La, Pr, Nd, Sm, Gd, Pb, U | | |
| BS-DISP active species | Si, S, K, Ca, Fe, Cu, La | | |
| Factors with BS mapping < 100% | Mixed (29%), Biomass Burning (98%) | Mixed (96%), Soil (99%) | Soil (56%), Haul Road Dust (97%), Mixed (99%) |
| DISP %dQ | 9.3E-5 | 0 | 1.8E-5 |
| DISP # of swaps | 0 | 0 | 0 |
| BS-DISP % of Cases Accepted | 87 | 97 | 85 |
| BS-DISP # of swaps | 10 | 0 | 12 |

8. ***The calibration with metals standards showed good accuracy (Table S1) after the Xact returned to the laboratory after field campaign. Was there anything done to fix the Pd signal drop during the field campaign? If so, what was cause of the problem and what was the fix? If not, did the instrument somehow fix by itself sometime between 8/25 and laboratory tests?***

Many parts including high voltage power supply (HVPS), x-ray tube, and filament cables had to be checked. We finally found a loose wire connection from the National Instruments AD converter and suspected that the position of the Pd rod changed slightly because of vibration during shipping. After the field campaign the HVPS had to be replaced. It appears that shipping can be tough on this instrument.

9. ***Page 11 Line 17-19. Please verify if No. 4 boiler fuels were used in upgrading facilities. From an earlier publication on oil sands stack emission (Wang et al., 2012), the boilers were fired with natural gas, process gas, and/or coke. Process gases (e.g., effluent from the sulfur recovery units) and coke, instead of No. 4 fuels, are likely the main sources of sulfur in that area. Careful survey and understanding of the industrial operations are required to reduce speculations.***

We agree and thank the reviewer. The comparison was simply to show an example of a mixed source combustion fuel, this section has been edited for accuracy and clarity as follows:

"The profile is suggestive of a mixed-combustion source (Lee et al., 2000; Van et al., 2008), such as coke, or the process gasses, which are comprised of effluent from the sulphur recovery units (Wang et al., 2012)."

10. ***Page 12. "It is speculated that this factor may have arisen from short term changes in fuel, such as a switch to oil combustion for heat/energy or the burning of coke. . ." This speculation of the source of "Upgrader Emission II" is not supported by hard evidence.***

We agree and had worked this text accordingly so as to emphasise that this was speculation and that this was raising possibilities rather than making any definitive claim

For clarity, this has been changed to: "It is speculated that this factor may have been due to two different stacks from within the upgrading process/facility. A less likely, but more fortuitous, possibility is that the Upgrader Emissions II factor was due to a short-term change in upgrader fuel that occurred only during the intensive measurement campaign and not during the long-term campaign"

11. ***Sections 3.2.3 and 3.2.4. I am not convinced that the soil factor and haul road factor can be reliably separated based on elemental composition. The profiles Figs. 2-3 are very similar. The overburden itself is stable and is not a large source of fugitive dust (Wang et al., 2015b). Dust is emitted from dikes built by overburden when heavy haulers are travelling on them. Then the emissions are similar to the haul road. Emissions of haul road dust occur when heavy haulers are moving on the unpaved haul road and when wind speed is high. Since diesel-fueled heavy haulers are the largest emitters of NOx, I would expect the haul road dust to better correlate with NOx than the "Soil" factor. Large dust plumes are often observed from tailings beaches under high wind conditions. Also there are several unpaved roads around the sampling sites, which are large dust sources. As shown in Figure 6 and its related discussion, both Soil and***

> *Haul Road factors increased during the day with very similar temporal patterns and are not driven by wind speed. These indicate that these two factors are likely driven by road dust. A single "dust" factor instead two may be more appropriate.*

While there are overlaps in the elements within the haul road dust and soil factors, there are enough differences to separate them as two separate sources. In fact, the source profiles for the Soil and Haul Road Dust were only weakly correlated (uncentered r=0.129), as is stated in the paper. Figure 6 simply shows the averaged-daily trends, not the overall hour-by-hour trends that PMF used, which are seen in Figure S9. These overall temporal trends do show clear differences in the time series of the two factors. Further, the speculated sources of the Haul Road Dust and Soil factors are due to on and off-road transportation. As the overburden dump is comprised of the overburden (top soil) in the area, it would be expected that any off-road transportation, whether on the dikes made of the overburden dump, or undisturbed overburden (top soil), would kick up dust comprised of the same material as the overburden dump. However, dust kicked up by vehicles on a road would be expected to have a slightly different elemental composition than that of the dust kicked up by the off-road vehicles, thus the two factors. Further the two factors, particularly in the intensive campaign, were very stable. These two factors can be combined in the 4-factor solution, but the reproducibility of the solution was poor and there were more factor swaps (please refer to the uncertainty analysis as shown earlier). This stability, along with lack of correlation, justifies the separation into two factors.

> *12. Table 2. Have the correlations between PMF factors and CO been looked at? CO is an indicator of biomass burning and/or vehicle emissions and may offer added evidence to the PMF factors.*

While this would be a useful comparison, of the species available to us at the sites, as well as during the two campaigns, CO was not a gas that available to us.

> *13. Page 20 Line 16-17. It is not accurate to categorize the Soil and Haul Rod Dust factors as "two with the transportation of the bitumen-rich oil". This excludes on-road vehicle and other dust sources.*

We agree that "the transportation of bitumen-rich oil" does not entirely encapsulate all of the transportation. To account for this, the sentence has been adjusted to read as follows:

 "two with on and off-road transportation"

> *14. Figure 6 and S10. It might be useful to examine the diurnal variation each factor.*

While the Soil and Haul Road Dust Factors showed what appeared to be regular, diurnal trends (See Figure S9), the other three factors did not exhibit any patterns that resemble a daily trend. Taking daily averages of such episodic occurrences would not aid in the identification of the factor.

*Technical Details*

> 1. *Figs 1, 7, S14 etc. All maps should have a scale bar to infer distances.*

Agreed, and they have been added to the Figures.

**2. Figs. 2, 3, S5, and S6. Add a note to indicate if the bar and symbol refers to the left or right hand side of the Y-axis.**

Agreed. For clarity, the following has been added to the figure descriptions:

"Factor concentrations depicted as bars, percentages depicted as circles."

**3. Page 6 Line 4. I suggest changing "dirt" to "oil sands".**

While this wording was constructed in a way to describe what "oil sand" is, the paper has been adjusted as follows:

"Bitumen-rich dirt" has been converted to "Oil Sands"

**4. Page 6 Lines 7-10. Each oil sands mining facility has multiple stacks that are connected to different upgrading processes. For example, Syncrude has a main stack, a FGD stack, and several smaller stacks. Furthermore, most stacks are equipped with pollution control devices. Some particles, e.g., ammonium sulfate, are formed in the FGD process designed to remove SO2 (Wang et al., 2012). Rewrite this sentence to accurately reflect this information.**

It is agreed that while there are numerous stacks, our comparison was to the measurements of the main stack taken by (Landis et al., 2012). However, in order to better reflect the totality of what happens, the wording has been changed to the following:

"Once at the upgrading facility the bitumen is separated from the slurry in large settling vessels, after which it is upgraded into different hydrocarbon streams using steam, vacuum distillation, fluid cokers, and hydrocrackers, these processes produce aerosol particles which are emitted to the air through both the main stack (Landis et al., 2012), as well as numerous other secondary stacks. Within these stacks, some particles are directly emitted from the upgrading processes, while others are created by the pollution control devices installed within the stacks (Wang et al., 2012)."

**5. Page 6 Lines 10-13. Evaporative emissions from tailings ponds may be an importance source of VOCs and secondary particles, but may not be an important source for primary particles. Instead, windblown dust from tailings beaches and dikes is a significant particle source (Wang et al., 2015b).**

It is agreed that evaporative emissions is likely not a significant source of primary particles. As such, the sentence has been altered to read:

"Other known sources of particles are: the large fleets of on and off-road vehicles, dust re-suspended by mining activities, windblown dust from the tailings ponds and dikes (Wang et al., 2015), and dust re-suspended from open petroleum coke piles."

*6. Fig. S3b-S3d showed 9 metals. Where are other metals?*

In order to do a comparison, hourly measurements were turned into daily averages. If in the course of the day over 50% of the hourly measurements were below detection limit or if the average itself was below detection limit, this daily average data point was removed due to increase unreliability associated with below detection limit measurements. In some cases this removed all daily average measurements for a metal.

*7. Write "Eqn 1" in supplemental materials in an equation form.*

Equations have been denoted as Arabic numerals in parentheses as suggested by the journal guideline.

*8. There are two Table S4*

Agreed, in sections S.5 and S.6, the table numbers have been altered to be "Table S6 and Table S7"

*9. Page 10 Section 3.1 and Section 3.5. I don't think it is fair to compare the 90th percentile in the oil sands to the average city values. If such comparison should be done, the 90th percentile of city values should be used. Also the three monitoring stations are considered as near-source monitoring due to their close distance to mining facilities. It is not surprising that their elemental concentrations are higher than many cities, considering the abundance of dust in the region (Shotyk et al., 2016).*

We agree that the 90th percentile does provide a different comparison than a straight mean comparison table. The main reason for including this table was to provide some context for the levels of PM2.5 metals relative to other sites in Canada where this type of data is available, which all happen to be cities. The second reason is to identify which metals are likely related to activities in the region. To do this we looked at the 90th percentile values which are more likely representative of intermittent plumes, than day to day averages.

To increase the clarity of these points, we made the following changes:

Within Section 3.1:

In order to clarify that the only reason that the oil sands concentrations were compared to the cities alone was due to lack of Canadian trace element data for remote sites, "cities" has been changed to "sites" in many places in order to clarify the intent of the comparison. For example, the first sentence, which reads "Average metal concentrations from the filter data were compared to measurements taken by the NAPS program at seven different Canadian sites (Environment and Climte Change Canada, 2015) (Table S6), which, due to a lack of comparable rural or remote baseline data, corresponded to different Canadian cities."

In order to clarify the use of the 90th percentile as a means to predict which metals come from anthropogenic sources, the end of section 3.1 has been altered to read:

"In the oil sands, large swaths of forest are broken up by the occasional mine or upgrader. When the wind comes from one of these directions, particularly the upgraders, there is a noticeable difference in the air quality. To illustrate this large variability, the 90th percentile of the various elements was calculated and

analysed, as species that show a high degree of variability are more likely from these intermittent pollution sources. The results of this showed that the previously discussed elevated elements showed peaks indicating large variability. Additionally, at the highest peaks the concentrations of S, Ba, Br, and Mn also showed large increases, which indicates that in the Oil Sands, they are likely caused by anthropogenic sources."

Within Section 3.5:

The second and third sentences have been altered to read "In particular, the concentrations of Si, Ti, K, Fe, Ca, and Al were, on average, higher than those measured in major Canadian cities. At their highest peaks the concentrations of S, Ba, Br, and Mn also exhibited large increases in concentrations."

**10. Page 13 Line 17. "lower concentrations of Mn and Fe" should be "higher. . .".**

We agree that the "lower" should be changed to "higher". This has been changed.

The sentence in the "Haul Road Dust" Section now reads as "What differentiated this factor from the Soil factor were the higher concentrations of Mn and Fe, and Ca. (Fig. 2)."

**11. Page 20 Line 4. Incomplete sentence: including. . .**

We agree that this is an incomplete sentence. And the "including" has been dropped from that sentence.

The sentence in "Implications of the Source Identification for Element Species" section now reads "As these factors represented a combination of multiple sources, the individual sources causing elevation of these elements is still not known; this limitation may help direct further studies."

References

Bari, M., Kindzierski, W.B., Wallace, L.A., Wheeler, A.J., MacNeill, M., Héroux, M.-È.; Indoor and outdoor levels and sources of submicron particles (PM1) at homes in Edmonton, Canada, Environ. Sci. Technol. (49), 6419-6429, 2015.

Bari, M., and Kindzierski, W.: Ambient fine particulate matter (PM2.5) in Canadian oil sands communities: Levels, sources and potential human health risk, Sci. Total Environ., 828-838, 2017.

Howell, S. G., Clarke, A. D., Freitag, S., McNaughton, C. S., Kapustin, V., Brekovskikh, V., Jimenez, J.-L., and Cubison, M. J.: An airborne assessment of atmospheric particulate emissions from the processing of Athabasca oil sands, Atmos. Chem. Phys., 14, 5073-5087, doi:10.5194/acp-14-5073-2014, 2014.

Kindzierski, W. and Bari, M.: Long-term temporal trends and influence of criteria pollutants on regional air quality in Fort McKay, Alberta, in: Proceedings of the IASTED International Conference on Unconventional Oils and the Environment, Calgary, Alberta, July 4-6, 2011, 162-168, 2011.

Kindzierski, W. and Bari, M.: Air quality in Alberta oil sands community of Fort McKay compared to urban residential locations in Canada in 2009, in: Proceedings of the Canadian Society for Civil Engineers 2012, Leadership in Sustainable Infrastructure, Edmonton, AB, 974-983, 2012.

Landis, M., Patrick Pancras, J., Graney, J., White, E., Edgerton, E., Legge, A., & Percy, K. Source apportionment of ambient fine and coarse particulate matter at the Fort McKay community site, in the Athabasca Oil Sands Region, Alberta, Canada. Sci. Total Environ. , 584-585:105-117, 2017.

Percy, K., Hansen, M., and Dann, T.: Air quality in the Athabasca Oil Sands Region 2011. In Alberta Oil Sands: Energy, Industry, and the Environment, Elsevier Press, Amsterdam, The Netherlands, 47-91, 2012.

Shotyk, W., Bicalho, B., Cuss, C., Duke, M., Noernberg, T., Pelletier, R., Pelletier, R., Zaccone, C.: Dust is the dominant source of "heavy metals" to peat moss (Sphagnum fuscum) in the bogs of the Athabasca Bituminous Sands region of northern Alberta, Environ. Int., 92-92, 494-506, 2016.

Wang, X. L., Watson, J. G., Chow, J. C., Kohl, S. D., Chen, L.-W., Sodeman, D. A., Legge, A. H., Percy, K E.: Measurement of real-world stack emissions with a dilution sampling system. In Alberta Oil Sands: Energy, Industry, and the Environment, Elsevier Press, Amsterdam, The Netherlands, 171-912, 2012.

Wang, X. L., Chow, J., Kohl, S., Yatavelli, L., Percy, K., Legge, A., and Watson, J.: Wind erosion potential for fugitive dust sources in the Athabasca Oil Sands Region, Aeolian Res., 18, 121-134, 2015.

---

## Author Comment (AC2) · 27 Jun 2017

*M. A. Bari*

*mdaynul@ualberta.ca*

*In this manuscript the authors investigated sources of ambient concentrations of elements in fine particulate matter (PM2.5) at three industrial locations in the Athabasca Oil Sands Region (AOSR) using 24-h (Dec. 2010 – Nov. 2012) and 1-h (August 2013) data. The receptor model EPA PMF3.0 was applied and seven emission sources were identified. In general, the results appear to be impressive and interesting for the international scientific community. However, I would like to raise some points that would be needed to address to better understand and reveal the sources of PM2.5 in the AOSR.*

*Some assumptions and interpretations have been made particularly in the methodology and result sections, which make the findings more uncertain. I would therefore suggest that the authors should consider major revisions as outlined in the specific comments.*

> *Specific comments:*
>
> *1. The authors investigated sources of PM2.5 using trace element concentrations that accounted for only a small fraction of PM2.5 mass. The authors should consider using all available chemical components e.g., cations (K+, Ca2+), anions (SO42–, NO3–). It is reported that additional chemical composition data would be available in the follow-up analysis (page 20). To the best of my knowledge, carbonaceous aerosol (EC/OC) measurements were not performed at oil sands region. Therefore, checking PM2.5 mass closure is helpful to identify the proportion of unaccounted mass, which can be included as an input variable (missing mass) in the model as suggested by Larson et al (2006) and have been applied in several other studies (e.g., Wu et al., 2007; Bari and Kindzierski, 2017). This helps to better explain some source factors.*

We agree that PM2.5 source apportionment using comprehensive chemical speciation data including major inorganic/organic aerosol will provide a better insight into PM2.5 sources in terms of the quantification of PM2.5 source contributions. However, as stated in the study objectives, we focused on the source identification of trace metal elements related to short-term sporadic events using high-time resolution data during the intensive campaign period. Furthermore, long-term filter data were used to evaluate the trace metal source apportionment from the intensive hourly measurements. However, only the trace and not the major component speciation data was available for the filter data. This the PMF analysis was done for the trace element data only so as to allow direct comparison between the findings form the two methods. Thus, while we agree with the reviewer's suggestion, it is beyond the possible scope of this current study and we may need a further study as stated in the manuscript"

"More generally, the elements used to create the factor profiles and thereby identify sources accounted for only a small portion of the total PM2.5 mass. This limitation will be addressed in follow-up analysis combining the Xact data with other concurrent, time-resolved, measurements of non-refractory components. Combining these data will provide a more complete mass reconstruction so as to allow apportionment of PM2.5 and further sources may be revealed by leveraging the perspective given by the additional composition information."

*2. It is not clear how the authors come up with the 5-factor solution using EPA PMF3.0. The authors provided justification for choosing the optimum number of factors screening basic criteria e.g., Q-values, G-space plots, Fpeak values. However, they didn't report any error estimation techniques such as bootstrapping (BS) analysis. The authors should apply the current version of the PMF model (EPA PMF5.0) that allows to better assess the uncertainty of PMF solutions, using three error estimation methods i.e., BS, displacement (DISP) and bootstrapping with displacement (BS-DISP) analysis.*

In the revised manuscript, EPAPMF 5 was used to calculate the error estimates of the PMF solution. Possible solutions were compared to determine the best solution based on their stabilities and uncertainties. Detailed information about the solution evaluation has been added in supplementary. Please find our responses to the reviewer #1.

*3. The selection of chemical elements for PMF analysis was based on the frequency of detection and species only >10% of the measurements above the detection limit were chosen. This may increase more uncertainty in PMF-resolved sources. It is suggested to exclude the elements with more than 70% of samples below the detection limit. The authors should include data quality information (including percentage of detection, below detected and missing values) in the supplemental. It is also suggested to provide QA/QC for laboratory analysis.*

Since air quality in the monitoring areas were frequently influenced by short-term episodic events lasting for several hours, we included  trace elements containing a high percentage of below the detection limit data to identify the short-term sources, that exhibited strong, plume-like behaviour. Compared to integrated filter samples, hourly measurements are advantageous to detect the sporadic events and below detection limit data can be useful to identify local sources. Furthermore, below detection limit data were down-weighted based on their signal-to-noise ratios to minimize modeling errors. While most elements that fulfilled this requirement had over 10% above detection limit data, a couple of metals had less than this, however, when these metals  were above the detection limit, they  were significantly higherfor several hours, suggesting a plume.

As suggested, we have added the percentage of data below the detection limit and missing data in Table S2, and clarified the description as follows.

"or data that exhibited strong, "plume-like" behaviour when it was above detection limit,"

All NAPS filter samples across the sites were routinely maintained and analyzed by Environment Canada in Ottawa. We briefly described the  filter sampling and analytical method in the manuscript as more detailed information is available elsewhere (Celo et al., 2011; Dabek-Zlotorzynska et al., 2011). A comparison of overlapping elements (i.e., Al, Ti, V, Mn, Fe, and Zn) measured by both ICP-MS and ED-XRF has been added in the supplementary (Please find our responses to the reviewer #1). The correlation coefficients ($r^2$) ranged from 0.81 to 0.96 with good agreements.

Celo, V., Dabek-Zlotorzynska, E., Zhao, J., Okonskaia, I., and Bowman, D.: An improved method for determination of lanthanoids in environmental samples by inductively coupled plasma mass spectrometry with high matrix introduction system, Anal. Chim. Acta., 706, 89-96, 2011.

Dabek-Zlotorzynska, E., Dann, T.F., Martinelango, P.K., Celo, V., Brook, J.R., Mathieu, D., Ding, L.Y., Austin, C.C., 2011. Canadian National Air Pollution Surveillance (NAPS) PM2.5 speciation program: methodology and PM2.5 chemical composition for the years 2003-2008. Atmospheric Environment 45, 673-686.

*4. The authors identified seven sources including two types of upgrader emissions, soil, haul road dust, biomass burning and two sources of mixed origin. It is suggested to improve the interpretation for describing some specific sources that are related to oil sands development. For example, adding secondary ions (SO42–, NO3–) in PMF analysis will help to better characterize the input of oil sands emissions.*

As discussed previously, the main objective of the study is to identify sources related to the increase of trace metal species. Since organic aerosol can be the largest PM contributor in this area (~70% of total PM2.5 mass were unidentified as shown in Bari and Kindzierski 2017), a follow-up source apportionment study including high resolution organic species measured by aerosol mass spectrometry would provide a better estimation of source contributions.

*5. Recent studies in the AOSR indicated fugitive dust (e.g., from oil sands tailings, unpaved roads and hauling fleet emissions) as a dominant source contributing ∼20%– 40% to PM2.5 (Landis et al., 2017, 2012; Bari and Kindzierski, 2017). The authors should check 'soil' and 'haul road dust' factors to better interpret the influence of fugitive dust.*

We agreed that the two "crustal" factors found in this study, soil and haul road dust, likely encompass a large portion of the "fugitive dust". As combined, our soil and dust factors make up between 31 and 38 % of the total metal element mass, this would agree. However, it should be noted that this is soil and dust factor contribution to the total metal mass in PM2.5 and care must be taken to compare with PM2.5 source contributions as shown in the references (Landis et al., 2017, 2012; Bari and Kindzierski, 2017).

Because the term "fugitive dust" encapsulates so much of the different sources of dust (such as from the oil sands tailings, unpaved roads (or off-road vehicles), and Hauling Fleet Emissions (or on-road vehicles), it makes more sense to keep them separate, as they describe two parts of the overall fugitive dust.

*6. The authors tried to compare the observed levels of PM2.5 elements in the industrial locations in the AOSR with other Canadian cities. Elemental levels at oil sands communities (e.g., Fort McKay and Fort McMurray) were not investigated here. Due to the unique nature of emission sources (not available in other Canadian region), the comparison analysis may not be informative and therefore suggested to exclude from the manuscript.*

We agree that in order to get a full idea of how the element levels in the Oil Sands compares to other areas, the concentrations would have ideally be compared to baseline rural and remote measurement sites with no industrial activity. However, we wanted to compare the oil sands data to sites in Canada where

this type of information has been measured, which happens to be the sites included, within the cities. Despite this limitation, a comparison to cities, urban sites known to be more polluted, some even with other industrial emissions nearby, do provide some context for the levels observed.

Further, while the measurements taken in the long-term study do not include Fort McMurray, one of the sites, AMS 13, is located in the south of Fort McKay, and would thus is representative of that community. Further the three sites do cover a range of the area, so  their average  provides an indication of how the area as a whole compares to other Canadian sites.

The paper has been adjusted to refer to the comparison cities as "sites".

---

## Author Comment (AC3) · 27 Jun 2017

*The manuscript by Phillips-Smith et al. investigated sources of PM2.5 in the Athabasca Oil Sands region based on PMF analyses of particulate metals. The metal species were derived from a long-term campaign in which filter-based measurements were conducted at three sites, and also from an intensive campaign in which semi-continuous measurements were performed at one of the long-term sites. Interestingly, PMF results were compared between these two campaigns. The topic of this manuscript is within the scope of the special issue "Atmospheric emissions from oil sands development and their transport, transformation and deposition". However, I cannot support its publication in the current form. As can be seen from my detailed comments given below, my major concerns are about the PMF results.*

1. *Page 3, line 25-28. It is not appropriate to list "modelling", "airborne studies" and "comparison of PM2.5 concentrations" successively in one sentence.*

We thank the reviewer for their feedback. In order to provide a brief overview of work completed on the topic of PM2.5 in the Oils Sands Region, the examples were simply listed. To better structure this, this section has been reworded as follows:

"Past research on $PM_{2.5}$ within the Athabasca Region has included overall and comparative emission and air quality analyses (Kindzierski and Bari, 2011; Kindzierski and Bari, 2012; Wang et al., 2012; Percy et al., 2012; Howell et al., 2014; Wang et al., 2015; Landis et al., 2017). Further studies have delved into modelling the emission sources through both computer-based (Cho et al., 2012) and measurement-based methods (Landis et al., 2012)."

2. *Page 4, line 9-13. The authors implied that no source apportionment study had been performed for the oil sands region using metal species of PM2.5. However, it is unclear whether there are any previous source apportionment studies using other PM2.5 components (e.g., water-soluble ions, organic carbon, elemental carbon and etc.). Please clarify.*

It is true that there has been another PMF study done in the Oil Sands, based on PM2.5 (Bari and Kindzierski, 2017). However, it did not focus on the metal analysis, or use high time resolution data to compare and contrast the results.

The section of the paper has been adjusted as follows:

"In past receptor modeling, open pit mining, upgrading, and fugitive dust have been identified as major emission factors in the oil sands region (Landis et al., 2012; Bari and Kindzierski, 2017). However, these emission factors were identified based solely on long-term, low time-resolution data."

3. *Page 4, line 24-26. No content in the results and discussion section corresponds to the second purpose presented here.*

In order to reduce the length of the paper, findings relating to the accuracy, precision, and consistency of the XactTM 625 instrument are introduced in section 2.5 and described in detail in the supplementary.

4. ***Line 29 in Page 4 to line 8 in page 5. This paragraph should be presented much more briefly, since all the descriptions involved here are repeated in the methods section.***

The text has been shortened by removing details related to the method as follows:

"Since December 2010, under the Enhanced Deposition Component of the Joint Canada-Alberta Implementation Plan for Oil Sands Monitoring (JOSM) Program, 24-hr integrated filter samples have been collected by Environment and Climate Change Canada in $PM_{2.5}$ at three sites (Fig. 1) operated by the Wood Buffalo Environmental Association (WBEA). As part of a 2013 summer intensive field campaign, hourly measurements were also made at one of the sites (Fort McKay South, AMS13) for one month (Aug. 10- Sept. 10) using a semi-continuous metal monitoring system."

5. ***Page 9, line 6-9. Please provide (representative) scatter plots comparing ICP-MS and ED-XRF measurement results on the same metals.***

Scatter plots for Al, Ti, V, Mn, Fe, and Zn measured by ICP-MS and ED-XRF have been added in the supplementary (Figure S3f). The correlation coefficients ($r^2$) ranged from 0.81 to 0.96 with good agreements. Please find our responses to the reviewer #1.

6. ***Page 11, line 12-15. It is unclear which PMF profile (i.e., Upgrader Emissions I shown in Figure 2 or 3) was used for the comparison to the profile reported by Landis et al. (2012). In addition, it is quite surprising that the regression analysis could show an r value of 1.00. Does this mean that the two profiles are exactly the same?***

In order to confirm that both campaigns identified the same factor, both profiles were compared to the same upgrader emission profile published by Landis et al. (2012).

The intensive campaign is referenced Figure 2, while the long-term campaign is referenced Figure 3.

Further, while surprising, both profiles so closely resembled the reported profile of Landis et al. (2012), that to 2 significant digits, their r values reported as 1.00. In the revised manuscript, uncentered correlation coefficient has been used as an unbiased metric to evaluate the level of similarity between the profiles of sources. This metric is particularly useful as it takes into account the similarity of minor peaks. Spearman ranked correlation analysis was performed on the comparison of temporal variations (i.e., time series).

The paper has been adjusted to clarify which correlation relates to which figure as follows:

In Section 2.4.2., "Spearman ranked correlation analysis was performed on the comparison of temporal variations, whereas uncentered correlation coefficient was used to evaluate the level of similarity between factor profiles."

In Section 3.2.1., "This factor was attributed to typical emissions from the upgrading processes based on the correlation (uncentered r=1.00 for the intensive campaign (Figure 2): uncentered r=1.00 for the longterm campaign (Figure 3)) of its elemental profile with an average profile derived from samples of $PM_{2.5}$ taken from main upgrader stacks in the area (Landis et al., 2012).”

“There were strong correlations in: i) the PMF factor profiles derived from the two methodologies and ii) the time series between the co-measured Xact and filter data of this factor (profile (uncentered r=1.00); time series (Spearman r=0.74, p<0.01)).”

7. *Page 11, line 29-30. V and Ni were used to indicate oil combustion. However, as shown in Figure 2, the majority of Ni was attributed to the Mixed Sources factor; on the other hand, negligible V was seen in the Mixed Sources factor. These results mean that the major sources of V and Ni are different. Consequently, I don't think it is reliable to attribute the Upgrader Emissions II factor to oil combustion, unless the authors could demonstrate that the V to Ni ratio calculated for this factor was comparable to that measured in source emissions from oil combustion.*

We agree that while the Upgrader II factor does not contain the highest percentage of Ni of all the factors, it does contain a significant amount (around 30% of the total). Further, the ratio of V to Ni for this factor is 5.5, which is very similar to the known ratio for heavy oil combustion (V/Ni = 5-7) (Huffman et al., 2000).

This section has been adjusted to account for this as follows:

“More specifically, this factor was attributed to oil or bitumen based fuel combustion because of the higher percentages of V and Ni, (Fig. 2), which are typical of oil combustion (Huffman et al., 2000; Lee et al., 2000). On average, the ratio of V to Ni in this factor profile was 5.5, which was comparable to heavy oil combustion with high sulphur contents reported by Huffman et al., 2000 (V/Ni=5-7).”

8. *Page 13, line 17. According to Figure 2 and 3, concentrations of Mn and Fe were higher in the Haul Road Dust factor compared to the Soil factor.*

We agree that the “lower” should be changed to “higher”. This has been changed to agree.

The sentence in the “Haul Road Dust” Section now reads as “What differentiated this factor from the Soil factor were the higher concentrations of Mn, Fe, and Ca (Fig. 2).”

9. *Page 13, section 3.2.5. Figure 3 indicates that biomass burning was the major source of Cd. Moreover, the biomass burning contribution to Cd (~80%) was more significant than its contribution to K (~60%). However, previous source emission studies typically suggest that biomass burning is not a strong source of Cd (e.g., Schmidl et al., Atmos. Environ., 42, 126-141, 2008 and references therein). The authors are required to provide references to support their discussions here, i.e., biomass burning could be the major source of Cd.*

Since the correlation between K and Cd was strong (r=0.74, w/o an outlier on Feb 9, 2012), Cd cannot be separated from the biomass factor. Although the high loading of Cd in biomass burning is not commonly found in other areas, source apportionment studies in Edmonton, Alberta reported the presence of Cd in biomass burning factors including our previous PMF study in Edmonton (~20% of total mass, Jeong et

al., 2011). Kindzierski and Bari (2015) and Bari et al., (2015) also found the high loading of Cd (37%-62%) as marker species of biomass burning in this area.

This section has been altered to address the above as follows:

"All of these elements, to different degrees, have been associated with different types of biomass burning in this (Jeong, et al., 2011; Kindzierski and Bari, 2015; Bari et al., 2015) and other regions (Van et al., 2008; Vassura et al., 2014; Alves et al., 2011)."

**10. Page 14, section 3.2.6. High abundance of Ni observed in the Mixed Sources factor should be discussed.**

Due to the nature this factor, we have concluded that it is from a mixture of sources, for which there was not sufficient data to resolve into their own, separate sources.

Regardless, the largest known emitter of Ni, at 2.5 tonnes in 2013, according to the 2013 NPRI data from near Wood Buffalo, is the Mildred Lake Plant Site of Syncrude. This facility also co-emitted Cu, Cr, Zn, and Se, which were more of the marker elements of the Mixed Sources Factor.

This section has been adjusted to reflect this information as follows:

Addition of "Ni" into the marker elements descriptions of the two campaigns for this factor.

"as well as Ni, K and Se, which suggested the inclusion of biomass, oil, or coal burning, perhaps the burning of scrap-brush"

Addition of the following sentence:

"Other elements within these Mixed Sources could have been the result of further industrial activity at the different plant sites, which are known to be large emitters of elements such as Ni, Cu, Cr, Zn, and Se (Environment and Climate Change Canada, 2015)."

References

Bari, M., Kindzierski, W.B., Wallace, L.A., Wheeler, A.J., MacNeill, M., Héroux, M.-È.; Indoor and outdoor levels and sources of submicron particles (PM1) at homes in Edmonton, Canada, Environ. Sci. Technol. (49), 6419-6429, 2015.

Bari, M., and Kindzierski, W.: Ambient fine particulate matter (PM2.5) in Canadian oil sands communities: Levels, sources and potential human health risk, Sci. Total Environ., 828-838, 2017.

Howell, S. G., Clarke, A. D., Freitag, S., McNaughton, C. S., Kapustin, V., Brekovskikh, V., Jimenez, J.-L., and Cubison, M. J.: An airborne assessment of atmospheric particulate emissions from the processing of Athabasca oil sands, Atmos. Chem. Phys., 14, 5073-5087, doi:10.5194/acp-14-5073-2014, 2014.

Jeong, C.-H., McGuire, M. L., Herod, D., Dann, T., Dabek-Zlotorzynska, E., Wang, D., Ding, L., Celo, V.,Mathieu, D., and Evans, G. J.: Identification of sources of PM2.5 at five urban sites across Canada, Atmos. Poll. Res., 2, 158-171, 2011.

Kindzierski, W. and Bari, M.: Long-term temporal trends and influence of criteria pollutants on regional air quality in Fort McKay, Alberta, in: Proceedings of the IASTED International Conference on Unconventional Oils and the Environment, Calgary, Alberta, July 4-6, 2011, 162-168, 2011.

Kindzierski, W. and Bari, M.: Air quality in Alberta oil sands community of Fort McKay compared to urban residential locations in Canada in 2009, in: Proceedings of the Canadian Society for Civil Engineers 2012, Leadership in Sustainable Infrastructure, Edmonton, AB, 974-983, 2012.

Kindzierski, W., and Bari, M.: Investigation of fine particulate matter characteristics and sources in Edmonton, Alberta. Final Report. University of Alberta, Edmonton, Alberta. 2015.

Landis, M., Patrick Pancras, J., Graney, J., White, E., Edgerton, E., Legge, A., & Percy, K. Source apportionment of ambient fine and coarse particulate matter at the Fort McKay community site, in the Athabasca Oil Sands Region, Alberta, Canada. Sci. Total Environ. , 584-585:105-117, 2017.

Percy, K., Hansen, M., and Dann, T.: Air quality in the Athabasca Oil Sands Region 2011. In Alberta Oil Sands: Energy, Industry, and the Environment, Elsevier Press, Amsterdam, The Netherlands, 47-91, 2012.

Wang, X. L., Chow, J., Kohl, S., Yatavelli, L., Percy, K., Legge, A., and Watson, J.: Wind erosion potential for fugitive dust sources in the Athabasca Oil Sands Region, Aeolian Res., 18, 121-134, 2015.

Wang, X. L., Watson, J. G., Chow, J. C., Kohl, S. D., Chen, L.-W., Sodeman, D. A., Legge, A. H., Percy, K E.: Measurement of real-world stack emissions with a dilution sampling system. In Alberta Oil Sands: Energy, Industry, and the Environment, Elsevier Press, Amsterdam, The Netherlands, 171-912, 2012.